# Words are all you need? Language as an approximation for human similarity judgments

**Raja Marjieh**[1,*], **Pol van Rijn**[2,*], **Ilia Sucholutsky**[3,*], **Theodore R. Sumers**[3], **Harin Lee**[2,4],

**Thomas L. Griffiths**[1,3,**], **Nori Jacoby**[2,**]

[*]/[**]**Equal contribution.**
[1]Department of Psychology, Princeton University
[2]Max Planck Institute for Empirical Aesthetics
[3]Department of Computer Science, Princeton University
[4]Max Planck Institute for Cognitive and Brain Sciences

## Abstract

Human similarity judgments are a powerful supervision signal for machine learning applications based on techniques such as contrastive learning, information retrieval, and model alignment, but classical methods for collecting human similarity judgments are too expensive to be used at scale. Recent methods propose using pre-trained deep neural networks (DNNs) to approximate human similarity, but pre-trained DNNs may not be available for certain domains (e.g., medical images, low-resource languages) and their performance in approximating human similarity has not been extensively tested. We conducted an evaluation of 611 pre-trained models across three domains – images, audio, video – and found that there is a large gap in performance between human similarity judgments and pre-trained DNNs. To address this gap, we propose a new class of similarity approximation methods based on language. To collect the language data required by these new methods, we also developed and validated a novel adaptive tag collection pipeline. We find that our proposed language-based methods are significantly cheaper, in the number of human judgments, than classical methods, but still improve performance over the DNN-based methods. Finally, we also develop 'stacked' methods that combine language embeddings with DNN embeddings, and find that these consistently provide the best approximations for human similarity across all three of our modalities. Based on the results of this comprehensive study, we provide a concise guide for researchers interested in collecting or approximating human similarity data. To accompany this guide, we also release all of the similarity and language data, a total of 206,339 human judgments, that we collected in our experiments, along with a detailed breakdown of all modeling results.

## 1 Introduction

Similarity judgments have long been used as a tool for studying human representations, both in cognitive science (Shepard, 1980; 1987; Tversky, 1977; Tenenbaum & Griffiths, 2001), as well as in neuroscience, as exemplified by the rich literature on representational similarity between humans and machines (Schrimpf et al., 2020; Kell et al., 2018; Linsley et al., 2017; Langlois et al., 2021; Yamins et al., 2014) whereby similarity patterns of brain activity are compared to those arising from a model of interest. Recent research in machine learning suggests that incorporating human similarity judgments in model training can play an important role in a variety of paradigms such as human alignment (Esling et al., 2018), contrastive learning (Khosla et al., 2020), information retrieval (Parekh et al., 2020), and natural language processing (Gao et al., 2021).

However, building a large dataset based on human similarity judgments is very expensive and often infeasible since the number of judgments required is quadratic in the number of stimuli – for $N$

---

[*]Correspondence: {raja.marjieh,is2961}@princeton.edu, pol.van-rijn@@ae.mpg.de

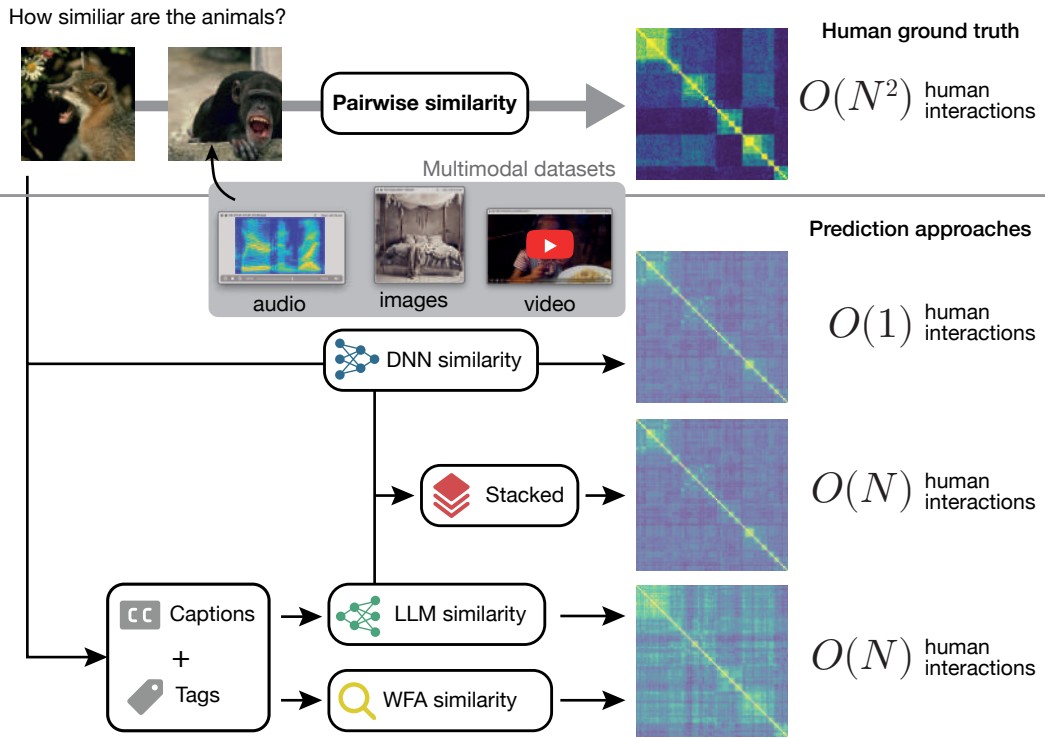

Figure 1: Comparing human similarity scores gathered through crowdsourcing with ML pipelines. We used data from three modalities: images, audio, and video. For each modality, we extracted deep model embeddings and gathered human captions and tags. Word- and language-embedding models, as well as simple word-frequency analysis, were used to predict human similarity judgments.

stimuli, $O(N^2)$ judgments are required[1]. For example, to fully quantify the similarity of all possible dyadic pairs of 50,000 images, one needs to collect on the order of 1.25 billion ($\sim \frac{50000^2}{2}$) human similarity judgments. Thus, human judgments are the main bottleneck for machine-learning methods based on similarity. For this reason, the majority of available human similarity datasets are small by machine learning standards (up to a few thousand objects).

Advancements in deep learning have brought an alternative approach that does not require extensive collection of human judgments. Specifically, the idea is to use the similarity between hidden representations in pre-trained deep neural networks (DNNs) to approximate human similarity (Peterson et al., 2018; Jha et al., 2020; Marjieh et al., 2022; Hebart et al., 2020; Roads & Love, 2021). Some of these methods also suggest fine-tuning representations on a small training set of human similarity judgments (Peterson et al., 2018). This, in turn, results in a significant reduction in the number of required human judgments down to $O(1)$ (given the pre-trained model). While such methods are promising, they still require access to strong pre-trained models which may not necessarily be available in all domains (e.g., medical datasets, niche modalities, low-resource languages, etc.). In addition, representations obtained from neural networks may not always overlap with human similarity representations, given that the models can be trained for different objectives (i.e., their embeddings may be poor approximations for human similarity).

A comprehensive comparison to assess which models perform well in predicting human similarity across different modalities is currently lacking in the literature. To this end, one of our main contributions in this paper is providing a first-of-its-kind large-scale evaluation of over 600 publicly-available pre-trained models as approximations for human similarity judgments on three modalities

---

[1]Depending on various assumptions, the full range of classical methods can require between $O(N \log N)$ (Jamieson & Nowak, 2011) and $O(N^3)$ (Hebart et al., 2020) human judgments. In this work, we used $O(N^2)$ human judgments (collecting all unique dyadic pairs) as the baseline for comparison

(images, audio, video). Our experiments reveal that there is a large gap in performance between the $O(1)$ DNN methods and the classical $O(N^2)$ similarity method we used as the baseline.

To address this gap, we propose a new class of $O(N)$ methods to efficiently and accurately approximate human similarity based on language. This is motivated by a long line of research in cognitive science suggesting that language is an extremely efficient way for humans to communicate information about their sensory environment (Murphy, 2004; Zaslavsky et al., 2018; Piantadosi et al., 2011; Jaeger & Levy, 2006). This in turn suggests that we can use textual descriptors to approximate similarity judgments across different modalities. Moreover, such textual descriptors can be collected at the cost of $O(N)$ human judgments (as people describe individual stimuli rather than pairs), which renders this method scalable.

We consider two approaches for approximating similarity from text data. One approach is to use pre-trained Large Language Models (LLM) to produce vector embeddings of the textual descriptions, and then use a measure of distance between these embeddings to approximate human similarity. This method is more domain-agnostic than the $O(1)$ deep learning methods as it only requires access to a pre-trained LLM regardless of the modality of the original dataset. However, there are some cases where the domain may be out-of-distribution for all available LLMs (e.g., niche technical fields), or where no LLMs are available at all (e.g., low-resource languages). In such cases, the other approach is to use Word-Frequency Analysis (WFA) methods from classical text processing literature (Barrios et al., 2016; Rouge, 2004; Beel et al., 2016),

As for the textual descriptions themselves, we consider two types, namely, free-text captions and concise word tags. Collecting captions for machine learning datasets is a well-established practice and can easily be done through crowdsourcing platforms. On the other hand, there is no consensus on best practices for collecting tags without a pre-existing taxonomy (i.e., open-set labels). To address this, we propose a novel adaptive tag mining pipeline called Sequential Transmission Evaluation Pipeline (STEP-Tag) which we describe in Section 2.2.4. As we will show, STEP-Tag allows to collect meaningful, diverse, and high-quality word tags for target stimuli in an online crowdsourcing environment.

Finally, we propose one additional set of hybrid approximation methods that combine sensory information with textual descriptions while still requiring $O(N)$ human judgments. For this approach, we propose to stack the embeddings derived from both domain-specific models (e.g., output from the last layer of an image classifier) with the LLM embedding of the respective textual description. When multi-modal models are available, we can similarly leverage the joint embedding of both the stimulus and its textual description.

We evaluate all of these novel and existing methods across multiple modalities. We test the relative contributions of linguistic and sensory information in approximating human similarity and show that our proposed language-based methods provide both accurate and efficient approximations across modalities, even though they do not require a trained modality-specific deep learning model. Crucially, with this large-scale evaluation, we are able for the first time to provide researchers with a comprehensive guide of the tools to use for approximating human similarity at scale.

To summarize, our contributions are as follows:

- We conduct a comprehensive comparison of human similarity approximation methods.

- We propose a novel modality-agnostic method for approximating similarity based on text and show that it is both efficient and competitive in terms of performance.

- We propose STEP-Tag, a novel adaptive tagging pipeline, and show that it is effective for crowdsourcing high-quality and diverse sets of word tags.

- We synthesize our findings into a detailed guide for researchers interested in approximating human similarity judgments at scale.

- We collect and release ground-truth and approximated versions of a large behavioral dataset ($N = 1{,}492$) across three different domains (images, audio, video), including two text-approximated similarity matrices for 1,000 audio clips and 1,000 video clips.

## 2 Datasets

### 2.1 Stimuli

Throughout this work, we considered five stimulus datasets across three different modalities – images, audio, and video – consisting of a total of 31,320 dyadic pairs labeled with similarity.

**Images** For images, we considered three datasets of common objects introduced in Peterson et al. (2018) – namely, animals, furniture, and vegetables – each consisting of 7,140 dyadic pairs (all unique pairs over 120 images).

**Audio** For audio, we used the RAVDESS corpus (Livingstone & Russo (2018), released under a CC Attribution license), which consists of semantically neutral sentences spoken by 24 US American actors to convey a specific target emotion. To construct a 1,000-recording subset, we selected 3 emotions per speaker per sentence. We randomly omitted 104 emotional stimuli and included all 96 neutral recordings (the dataset only contains 2 neutral recordings per speaker per sentence). To construct the subset composed of 4,950 dyadic pairs (all unique pairs over 100 recordings), we randomly selected ∼13 recordings per emotion from the 1,000.

**Video** Finally, for the video dataset, we considered the Mini-Kinetics-200 dataset (Xie et al., 2018) (released under a CC BY 4.0 International License), which contains a large set of short video clips of human activities from 200 activity classes. Specifically, we focused on the validation split, which contains 5,000 videos in total. To construct our 1,000-video dataset, we sampled 5 random videos from each of the 200 activity categories. The 100-video subset (4,950 dyadic pairs) used in the similarity judgment collection experiment was then generated by sampling 100 random stimuli from the 1,000 list.

### 2.2 Human judgment collection

#### 2.2.1 Participants

We collected data from $N = 1,492$ US participants for the new behavioral experiments reported in this paper. Participants were recruited anonymously from Amazon Mechanical Turk and provided informed consent under an approved protocol by either the Institutional Review Board (IRB) at Princeton University (application 10859) or the Max Planck Ethics Council (application 2021_42) before taking part. Participants earned 9-12 USD per hour, and each session lasted less than 30 minutes. To help recruit reliable participants, we required that participants are at least 18 years of age, reside in the United States and have participated in more than 5,000 previous tasks with a 99% approval rate (see Supplementary Section B for additional details about the behavioral experiments). All experiments were implemented with the Dallinger and PsyNet frameworks designed for automation of large-scale behavioral research (Harrison et al., 2020). In Supplementary Section A.1, we include the data that was collected, instructions used, and code for replication of the behavioral experiments. We also provide the code for computational experiments and analysis.

#### 2.2.2 Similarity judgments

We collected two batches of pairwise similarity judgements, one for each of the audio and video subsets, and were provided access to the similarity matrices for the three image datasets by the authors of Peterson et al. (2018). For each pair we collected ∼ 5 similarity judgments to average out inter-rater noise.

#### 2.2.3 Captions

We collected free-text captions for the video and audio datasets. Captions for the image datasets were already collected by Marjieh et al. (2022) and used here with permission. For each stimulus, we collected ∼ 10 captions.

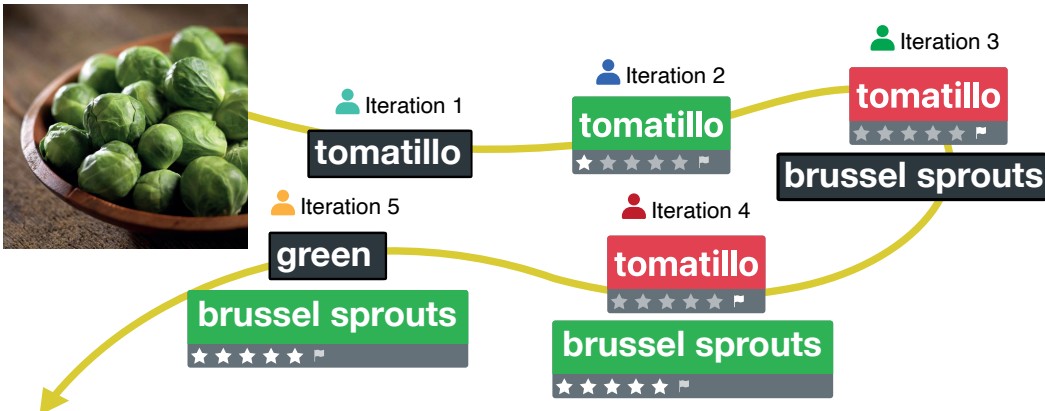

Figure 2: STEP-Tag, our novel tag-mining paradigm. We ran an adaptive process in which results of one iteration are used as inputs for subsequent iterations. In every iteration, participants can add a new tag, rate the relevance of existing tags or flag tags that are inappropriate.

### 2.2.4 TAGS

We propose a novel adaptive tag pipeline for simultaneous data collection and evaluation called Sequential Transmission Evaluation Pipeline (STEP) and apply it in the context of semantic tag mining (STEP-Tag). Our paradigm, STEP-Tag, allows researchers to efficiently collect high-quality word tags for a given stimulus (Figure 2) and extends existing crowdsourcing text-mining techniques (Von Ahn & Dabbish, 2008; 2004; Krishna et al., 2017; Law et al., 2007) by integrating ideas from transmission chain experiments (Kirby et al., 2008; Griffiths & Kalish, 2005). In STEP-Tag, participants adaptively create tags for a set of target stimuli and simultaneously evaluate the annotations made by previous participants. In each trial, participants are first given a stimulus (e.g., an image or audio fragment) and rate the relevance of tags that were created by other participants (on a 5-interval Likert scale) or flag a tag if they find it inappropriate (with tags removed if more than two people flag the tag). Next, participants are also given the opportunity to add new tags if they feel a relevant tag that describes the stimulus is missing. The results of the annotation procedure of one participant then propagate to the next participant (additional details about the paradigm, and screenshots are provided in Supplementary Section B.6). Ultimately, as the process unfolds over many iterations, meaningful tags are extracted and validated by multiple participants, enabling efficient open-label collection of a desired dataset.

To validate STEP-Tag, we compared it against several baselines: (i) randomly selecting only a single high-rated tag from the last iteration of STEP-Tag per stimulus, (ii) using tags only from the first iteration of STEP-Tag (equivalent to non-adaptive tag collection), and (iii) using class labels instead of tags. We found that tags produced after multiple iterations of STEP-Tag outperformed all three baselines in terms of quality (i.e., downstream performance for similarity reconstruction) and diversity (see Supplementary Section B.6.1).

## 3 MODELS

### 3.1 DNN-BASED METHODS

We tested a wide range of pre-trained ML models that do not rely on text (overall we tested 611 models) and compared their internal representations to human similarity judgments and text-based predictions (Figure 1A). We compiled our model pool by leveraging pre-trained model repositories (or zoos) available online. In particular, for images we use 569 pre-trained models from the `pytorch-image-models` package `timm` (Wightman, 2019), for audio we use 36 pre-trained models available in the `torchaudio` package (Yang et al., 2021) (see also Supplementary Figure 10 for an analysis of layer depth), and for video we use 6 pre-trained models available from the `PyTorchVideo` package (Fan et al., 2021). Because of the recent success of multimodal training, we additionally included 9 multimodal models based on CLIP from OpenAI's public implementa-

tion (`https://github.com/openai/CLIP`) for the image datasets, and compared them to "stacked" representations (i.e., concatenating embeddings from separate image and text models).

## 3.2 LLM-BASED METHODS

**Tags**   To embed tags we used ConceptNet Numberbatch (CNNB) which is a word-embedding model trained on the ConceptNet knowledge graph that leverages other popular word embedding models such as word2vec and GloVe (Speer et al., 2017). We experimented with several algorithms for computing similarity between sets (or multi-sets) of tags and share the details in Supplementary Section C.1.2. As a control, for images we also tried converting tags into a caption of the form "This is an image of tag1, tag2, . . ." and embedding them using a language model (see Supplementary Section C.1.2).

**Captions**   To embed captions, we used four pre-trained LLMs from HuggingFace (Wolf et al., 2020): 'bert-base-uncased', 'deberta-xlarge-mnli', 'sup-simcse-bert-base-uncased', and 'sup-simcse-roberta-large'. SimCSE is a pre-training procedure that uses semantic entailment in a contrastive learning objective (Gao et al., 2021). According to BERTScore (Zhang et al., 2020), the latter three models are ranked in the top 40 models in terms of correlation with human evaluations on certain tasks, with 'deberta-xlarge-mnli' ranked first. However, in our experiments, we found that embedding similarity computed from 'sup-simcse-roberta-large' has the highest correlation with human similarity judgments out of the four models. For SimCSE-based models, we used representations from the (final) embedding layer (where the SimCSE contrastive objective is actually applied). For the other two models, we computed embeddings from every layer, but restricted the main analysis to embeddings from the penultimate layers. This was done in order to be consistent with our procedure for DNNs.

**Other methods**   For the image datasets, we also considered several other methods that made use of LLMs but do not fit into the categories described above. One approach was using prompts with GPT3 (Brown et al., 2020) in a text-completion setup to directly predict similarity without extracting embeddings (see Supplementary Section C.1.3 for details). We also tried using pre-trained image captioning models to generate captions automatically (i.e. this would reduce $O(N)$ language-based methods to $O(1)$) but this resulted in poor performance (see Supplementary Section C.1.3 for details).

## 3.3 STACKING METHODS

We produce stacked representations for each modality by concatenating the single best-performing (see Figure 3) LLM's embeddings with the embeddings from the five best-performing DNNs into a single set of long embeddings. Since the two sets of embeddings come from different spaces, we add a single tunable hyperparameter for rescaling the LLM embeddings. This hyperparameter can be set manually, but we use a small number of ground-truth similarity judgments (we use dyadic pairs for just 20 stimuli) to optimize it automatically.

## 3.4 WORD FREQUENCY ANALYSIS (WFA) METHODS

The aim of the WFA methods is to enable similarity approximation from language using traditional embedding-free techniques. Such techniques are particularly useful for low-resource languages or cross-cultural comparisons (Cowen & Keltner, 2017; Barrett, 2020), for which pre-trained models are lacking, as they work solely on the basis of the text itself. The WFA methods we considered included measuring co-occurrence, Rouge score, bm25s, and tf-idf. We provide details on each of these procedures in Supplementary Section C.2.

## 3.5 PERFORMANCE METRIC

We quantified performance by computing the Pearson correlation $r$ between approximated similarity scores and the ground-truth human similarity scores for all the unique dyadic pairs in a dataset. We compared the performance of the different prediction methods to the inter-rater reliability (IRR) of participants, which serves as an approximate upper-bound on performance. Following Peterson et al. (2018), we computed IRR for each human similarity matrix using the split-half correlation method with a Spearman-Brown correction (Brown, 1910).

# 4 RESULTS

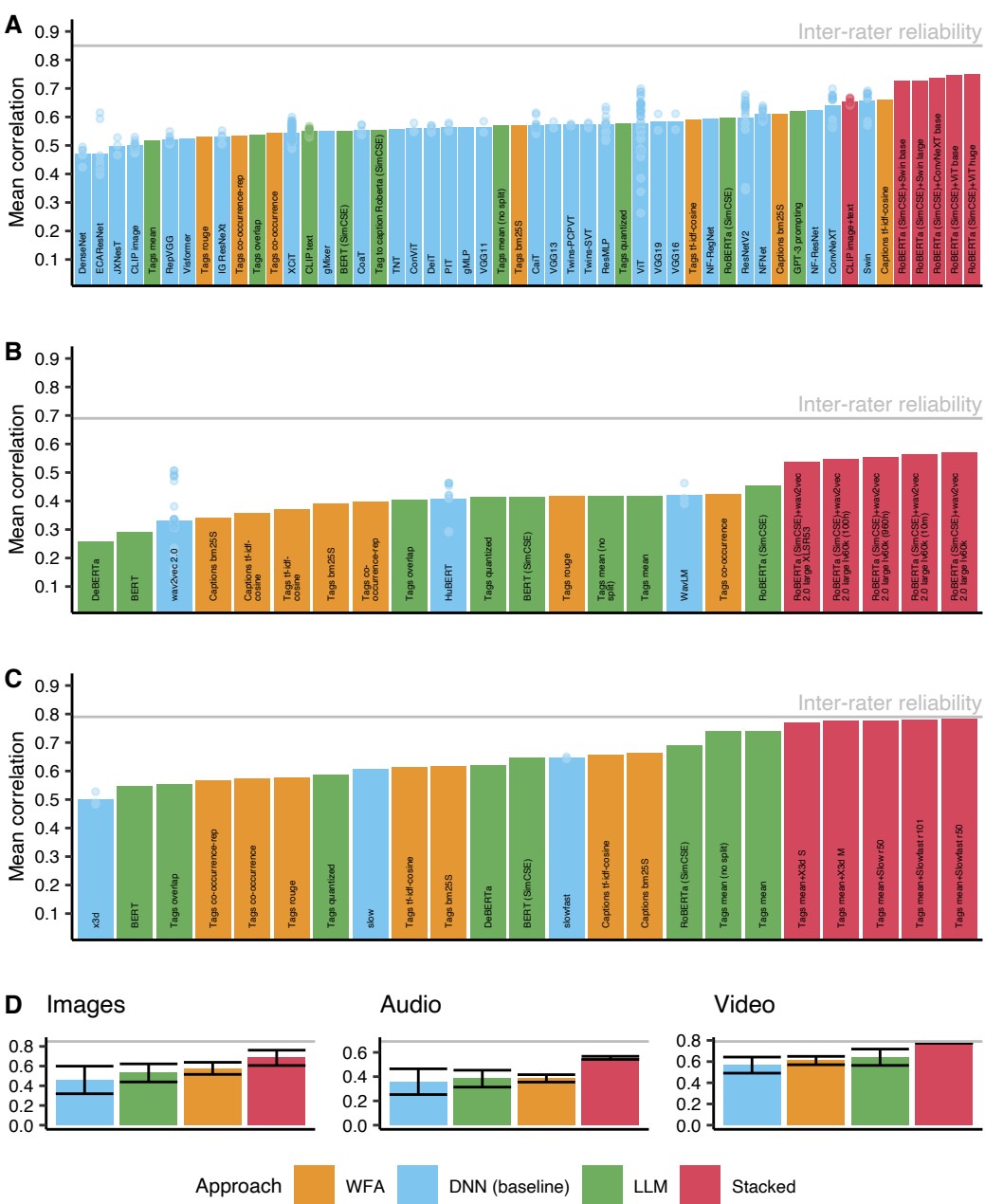

Figure 3: Correlation to human similarity. **A**: Top 50 models averaged over the 3 image datasets. **B**: Audio dataset. **C**: Video dataset. Each DNN baseline bar averages over multiple variants of the same architecture; the dots indicate average correlation of individual variants of the architecture. **D**: Average for each method type for each modality. The error bars are standard deviations.

Figure 3 summarizes the performance of the various techniques across the three modalities. Note that the image modality results in Figure 3A are averaged across the three image datasets and only show the top 50 methods for this modality due to space constraints. Figure 3D shows the mean performance of the methods of each type for each modality. When viewing these results, a clear hierarchy emerges. While no approximation methods can perfectly match the ground-truth pairwise similarity, (see the gap between the methods and IRR), stacked ones get close and are consistently more aligned with

human similarity than other methods across all three modalities. Text-based methods come next in this hierarchy, followed by DNN-based ones. We also considered supervised methods that reweight DNN-based embeddings based on a small set of human similarity judgments, but we found that the performance was unstable (see Supplementary Section C.3 for details).

The pre-eminence of stacked results suggests that LLMs and DNNs capture at least some different sources of variance in human similarity judgments. This is reinforced by our surprising finding that stacked representations from CLIP, a state-of-the-art jointly pre-trained multi-modal model, do not outperform stacked representations from independently trained models. We hypothesize that this happens because information is lost from both modalities when optimizing for a joint embedding. However, we note that the modest size of the performance gap between stacked and LLMs/DNNs, suggests that there is also significant overlap between aspects of human similarity captured by language and perception.

To investigate the effect of architecture and downstream task (e.g., classification) performance on alignment of DNNs with human similarity, for the image modality we compared similarity approximation performance against the number of model parameters on a log scale (Figure 4A) and ImageNet classification performance (Deng et al., 2009) (Figure 4B). Overall, we found a positive correlation between similarity approximation performance and the number of model parameters ($r = 0.39, p < 0.001$) and a smaller but still significant positive correlation with performance on ImageNet ($r = 0.26, p < 0.001$), There were some notable exceptions with particularly high ImageNet performance but low similarity performance, such as the image transformer BEiT (Bao et al., 2021).

Finally, we leverage both DNN-based methods and our proposed language-based methods to approximate similarity matrices that would otherwise require an unaffordable number of human similarity judgments to collect all dyadic pairs. Specifically, we approximate the two similarity matrices corresponding to all 1,000 audio clips and 1,000 video clips in our datasets using every method listed for each of those modalities in Figure 3. We provide visualizations of the resulting matrices at `https://words-are-all-you-need.s3.amazonaws.com/index.html`. We note that to exhaustively collect all dyadic pairs with five judgments per pair would normally require roughly 2.5 million human judgments for each of these matrices.

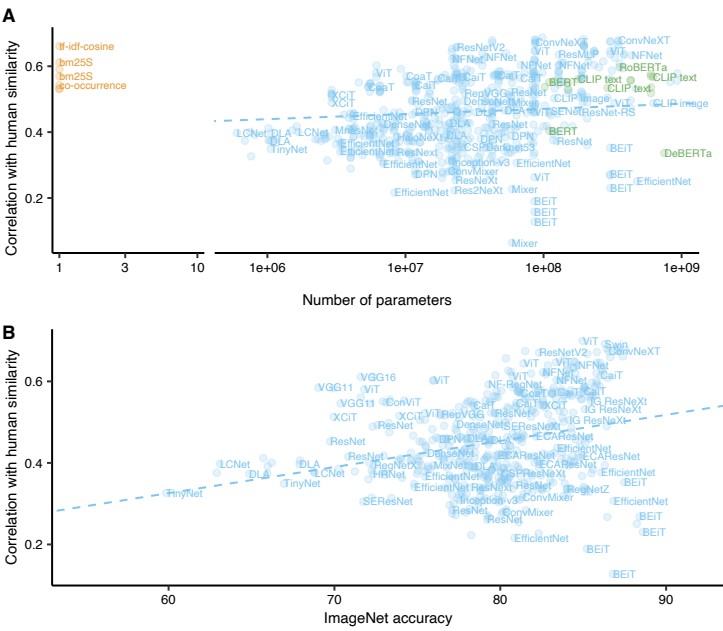

Figure 4: Correlation to human similarity judgments as a function of **A**: number of model parameters; and **B**: ImageNet accuracy.

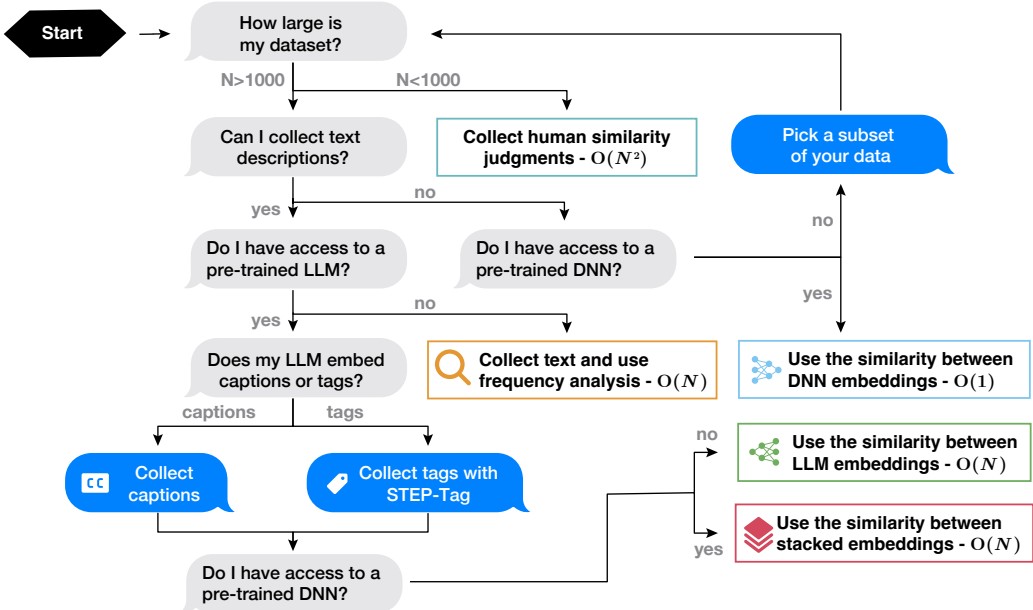

Figure 5: Guide to collecting and estimating human similarity judgments at scale.

## 5 DISCUSSION AND CONCLUSION

In this work, we compared novel and existing methods for approximating human similarity judgments. The main contributions can be summarized as follows: 1) we provide a simple and accessible approach for approximating $O(N^2)$ human similarity judgments using $O(N)$ annotations, 2) we propose a new adaptive pipeline STEP-tag for tag mining, 3) we evaluate our approach against 600+ domain-specific state-of-the-art DNNs, and 4) we publicly release all data comprising 206,339 human judgments.

Based on these, we are now able to provide researchers with a best-practices guide to collecting similarity datasets. Our guide is based on two bottlenecks that researchers may face: one is the limit on the number of judgments that can be collected (e.g., due to cost) and the second is the availability of pre-trained models (i.e., either DNNs or LLMs). Our results make it clear that deep learning can provide good approximations for human similarity. In fact, when both pre-trained LLMs and DNNs are available, stacking their representations is consistently the best approach. However, even when neither type of pre-trained models are available, we suggest that classical word-frequency analysis methods still provide researchers with an efficient and competitive method for approximating human similarity. Our guide, comprehensively covering these and other cases, is laid out in Figure 5.

One limitation of this work is that while similarity proxies generated from our pipeline can support ML datasets, they are also at risk of baking in high-level human biases that can lead to adverse societal implications, such as amplifying race and gender gaps. Researchers should devote utmost care to what they choose to incorporate in their training objective. Another limitation of our work is the fact that we were restricted to English text data and US participants. However, we believe that our approach and proposed methods (especially STEP-tag and the word-frequency methods) pave the way for the study of cross-cultural variation of human semantic representations by providing efficient tools for crowdsourcing high-quality semantic descriptors across languages. This is particularly relevant for low-resource languages, where our tag-mining techniques can work even with the absence of pre-trained ML models (Thompson et al., 2020; Barrett, 2020). We are currently expanding our work to include more languages and diverse cultures. Taken together, our results showcase how we can leverage language to make machine representations more human-like. Moreover, it highlights the importance of combining machine learning and cognitive science approaches for mutually advancing both fields. In particular, we believe that the methodologies adopted in this work have the potential to greatly advance basic research on naturalistic representations in cognitive science.

ACKNOWLEDGMENTS

This work was supported by a grant from the John Templeton Foundation to TLG, an NDSEG fellowship to TRS, and an NSERC fellowship (567554-2022) to IS.

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

SUPPLEMENTARY MATERIALS

## A  STIMULI AND DATA

### A.1  CODE AND DATA AVAILABILITY

A link is provided to the public, containing all the data collected for this project during the review stage.[2] It includes the new human behavioral data, the computational experiments with machine learning models, and all the necessary analyses scripts for producing the results. Additionally, the repository includes the Dallinger/PsyNet source codes for reproducing the behavioral experiments. Finally, we present an interactive visualization for exploring the similarity between stimuli as experienced by humans and different methods reported in the paper.[3]

## B  BEHAVIORAL PARADIGMS

### B.1  PARTICIPANTS

The exact number of participants for each of the 9 new behavioral experiments is reported in Table 1.

Table 1: Behavioral experiment summary table.

| Modality | Paradigm | Respect | Total stimuli | Trials per participant | Section | $N$ | Pre-screening |
|---|---|---|---|---|---|---|---|
| Images | Tags | Animals | 120 | 60 | 2.2.4 | 56 | LX |
| Images | Tags | Furniture | 120 | 60 | 2.2.4 | 58 | LX |
| Images | Tags | Vegetables | 120 | 60 | 2.2.4 | 57 | LX |
| Audio | Similarity | Emotions | 100 | 85 | 2.2.2 | 252 | HT |
| Audio | Captions | Emotions | 1,000 | 50 | 2.2.3 | 151 | HT, LX |
| Audio | Tags | Emotions | 1,000 | 50 | 2.2.4 | 217 | HT, LX |
| Video | Similarity | Activities | 100 | 85 | 2.2.2 | 284 | HT |
| Video | Captions | Activities | 1,000 | 50 | 2.2.3 | 196 | HT, LX |
| Video | Tags | Activities | 1,000 | 50 | 2.2.4 | 221 | HT, LX |

$Note.$ '$N$' denotes the number of participants included in the analysis; 'LX' denotes the LexTALE English proficiency pre-screening task; 'HT' denotes the headphone test.

### B.2  IMPLEMENTATION

All behavioral experiments were implemented using the Dallinger[4] and PsyNet (Harrison et al., 2020) frameworks. Dallinger is a modern tool for experiment hosting and deployment which automates the process of participant recruitment and compensation by integrating cloud-based services such as Heroku[5] with online crowd-sourcing platforms such as AMT. PsyNet is a novel experiment design framework that builds on Dallinger and allows for flexible specification of experiment timelines as well as providing support for a wide array of tasks across different modalities (visual, auditory and audio-visual). Participants interact with the experiment through their web-browser, which in turn communicates with a backend Python server responsible for the experiment logic.

### B.3  PRE-SCREENING

A common technique for filtering out participants that are likely to deliver low-quality responses, as well as automated scripts (bots), is to implement pre-screening tasks prior to the main part of

---

[2]**Code and data:** `https://osf.io/kzbr5/?view_only=3dea58e008ce41c290ef0f374bdbf444`
[3]**Interactive plots:** `https://words-are-all-you-need.s3.amazonaws.com/index.html`
[4]`https://dallinger.readthedocs.io/`
[5]`https://www.heroku.com/`

each experiment. Failing the pre-screening tasks results in early termination of the experiment. Nevertheless, participants are still compensated for their time regardless of whether they fail or succeed on a pre-screener to ensure fair compensation. The role of pre-screeners in our studies was to realize two main criteria for data quality, namely, a) to be able to collect high-quality text descriptors, and b) to ensure that participants are able to inspect the target stimuli properly (in particular the audio component in prosody and videos). To do this, we implemented two pre-screening tasks, an English proficiency test and a standardized headphone test (used only for audio and video experiments). Table 1 provides details on which pre-screeners were used in each of the behavioral experiments.

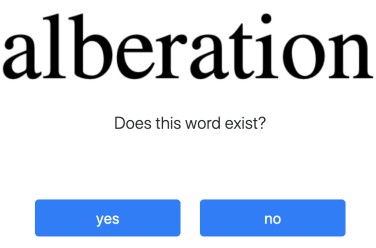

Figure 6: Example trial from the LexTALE pre-screening task (Lemhöfer & Broersma, 2012).

**English proficiency test**. To test participants' English proficiency, we used LexTALE, a lexical decision task developed in Lemhöfer & Broersma (2012). In each trial, participants were briefly presented (1 second) with either a real English word or a made up word that does not exist. Participants were instructed to guess whether the word was real or not. A total of 12 trials (half of them being real words) were presented, and 8 of them needed to be correct for the participant to pass. The presented words were: hasty, fray, stoutly, moonlit, scornful, unkempt, mensible, kilp, plaintively, crumper, plaudate, alberation. An example trial is shown in Figure 6.

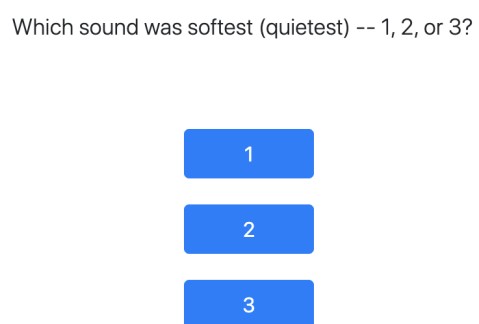

Figure 7: Example trial from the headphone pre-screening test (Woods et al., 2017).

**Headphone test**. We used the headphone test developed by Wood et al. (Woods et al., 2017), which is used as a standard pre-screener for high-quality auditory psychophysics data-collection procedures (Milne et al., 2021). The test is designed to ensure that the participants are wearing headphones and are able to perceive subtle differences in volume. The task consists of a forced choice task, in which three consecutive tones are played, and the participant has to identify which of them is the quietest. Crucially, these tones are constructed to exhibit a phase cancellation effect when not using headphones, and therefore making it difficult for non-headphone users to identify the quietest tone. Participants had to answer 4 out of 6 trials correctly to pass this test. An example trial is shown in Figure 7.

Figure 8: Screenshot from the similarity judgment task over video pairs.

## B.4 SIMILARITY JUDGMENTS

In the present work, we collected similarity judgments across audio and video datasets. Each dataset comprised of 4,950 unique pairs corresponding to the number of unordered subsets that contain two distinct objects (i.e., excluding self-similarity), within a set of 100 stimuli. We did not collect similarity judgments over the three datasets of images, as these were provided in Peterson et al. (2018) (and used here with permission). The experiments proceeded as follows: upon completion of the consent form and the pre-screening tasks, participants received instructions regarding the main experiment:

> **Audio.** In this experiment we are studying how people perceive emotions. In each round you will be presented with two different recordings and your task will be to simply judge how similar are the emotions of the speakers.

> **Video.** In this experiment we are studying how people perceive activities. In each round you will be presented with two different videos and your task will be to simply judge how similar are the activities in them.

The instructions then continued as follows:

> You will have seven response options, ranging from 0 ('Completely Dissimilar') to 6 ('Completely Similar'). Choose the one you think is most appropriate. Note: no prior expertise is required to complete this task, just choose what you intuitively think is the right answer.

> The quality of your responses will be automatically monitored, and you will receive a bonus at the end of the experiment in proportion to your quality score. The best way to achieve a high score is to concentrate and give each round your best attempt.

> The experiment will begin now. You will take up to 85 rounds where you have to answer this question. Remember to pay careful attention in order to get the best bonus!

As described in the instructions, in each trial, participants rated the similarity between a pair of sounds (how similar are the emotions of the two speakers?) or videos (how similar are the activities in the following two videos?) on a scale ranging from 0 (completely dissimilar) to 6 (completely similar) (Figure 8). Overall, participants completed 85 trials on a random subset of the possible pairs. To further motivate participants to provide good responses, we gave them an additional performance bonus for providing consistent data. Among the 85 trials, 5 trials were repeated for consistency checking. The responses were converted into a performance score by computing the Spearman correlation between the original and repeat ratings. Perfect scores resulted in a 10 cent bonus.

## B.5 CAPTIONS

We collected free-text captions for the video and audio datasets. Captions for the image datasets were previously collected in Marjieh et al. (2022) and used here with permission. After completing the consent form and pre-screening tests, participants received the following instructions:

> **Audio.** In this experiment we are studying how people describe emotions. You will be presented with different recordings of speakers and your task will be to describe their emotions. In doing so, please keep in mind the following instructions
>
> - Describe all the important aspects of the recording.

> **Video.** In this experiment we are studying how people describe activities in videos. You will be presented with different videos of activities and your task will be to describe their content. In doing so, please keep in mind the following instructions
>
> - Describe all the important activities in the video.

As well as the following guidelines adapted from Marjieh et al. (2022):

> - Do not start the sentences with "There is" or "There are".
> - Do not describe unimportant details.
> - You are not allowed to copy and paste descriptions.
> - Descriptions should contain at least 5 words.
> - Descriptions should contain at least 4 unique words.
>
> Note: No prior expertise is required to complete this task, just describe what you intuitively think is important as accurately as possible.

> The quality of your captions will be monitored automatically and providing low quality and repetitive responses could result in early termination of the experiment and hence a lower bonus.

> You will describe up to 50 recordings.

These guidelines were enforced to ensure that participants deliver sufficiently informative captions that are not repetitive. In each trial of the main experiment, participants described a single audio (please describe the emotions of the speaker) or video stimulus (please describe the activity in the video). Overall, participants described up to 50 randomly presented stimuli. To filter out bad participants that tend to deliver repeated responses, in each trial (excluding the first 4 trials) we computed the mean edit distance between their current response and all previous responses that they previously provided using the `partial_ratio` function in `thefuzz`[6] Python package for fuzzy string matching. This function returns for a pair of input strings a matching score between 0 and 100 (100 being identical strings). Early termination was enforced if the mean response matching score was above 80. The idea here was to prevent participants from copying and pasting the same response over and over again (or varying it only slightly).

## B.6 TAGS

For the image, audio, and video datasets, we collected tag data, i.e., concise labels that describe the salient features of a stimulus. To do so, we developed a novel tag mining paradigm called STEP-Tag in

---

[6]https://github.com/seatgeek/thefuzz

Figure 9: Screenshot of an example tag mining task for videos. The tag "picking" received 5 stars (very relevant), whereas the tag "apple" is flagged (marked as irrelevant).

| Dataset (# of stimuli) | mean | std | total |
|---|---|---|---|
| Vegetables (120) | 3.2 | 1.1 | 385 |
| Furniture (120) | 5.2 | 1.7 | 627 |
| Animals (120) | 8.2 | 2.7 | 988 |
| Audio-emotions (1000) | 9.1 | 3.5 | 9092 |
| Video-activities (1000) | 8.5 | 2.9 | 8482 |

Table 2: Mean, standard deviation, and total number of tags collected for each dataset.

which each stimulus was treated as a separate "chain" (see Figure 2 in the paper). When the stimulus was presented for the first time, the participant was asked to provide at least one tag. For the following iterations, we sequenced participants so that each of them had to rate the tags provided by participants from the previous iterations within the same chain. The rating was either choosing between one (not very relevant) to five stars (very relevant), or marking the tag as completely irrelevant by using the flag icon (see Figure 9). Participants could optionally introduce new tags that will subsequently be presented to other participants assigned to the same chain. Participants could only provide tags that were not already present, and they had to be in lower-case letters. To discourage frequent use of long word combinations, a pop-up window appeared if participants used two or more white spaces (i.e., three or more words) to warn that long combinations should only be used when completely necessary. This process continued for at least 10 iterations, after which we checked at each consequent iteration whether the chain was "full". We considered a chain to be full if its latest iteration had at least 2 tags that were rated at least 3 times and had a mean rating of 3 stars. If a chain was not full after 20 iterations, we stopped collecting further iterations. Since each experimental batch lasted for a fixed duration of less than one day, in some cases we did not complete all chains, and a few chains had fewer iterations (3 for vegetables, 6 for animals and 2 for furniture, out of 120 chains each). Our experiment incentivized participants to provide new tags by paying them a performance bonus of 0.01 USD for every up-vote (i.e., not flagged) given by other participants. On the contrary, if two or more tags of the same participant were flagged by others, the participant was excluded (the participant received a warning after the first flag). We provide summary statistics on the number of collected tags in Table 2.

After accepting the consent form and passing the pre-screening tasks, participants received introductory instructions regarding the main experiment:

> **Images**. Rate & Tag animals/furniture/vegetables! Thanks for participating in this game! In this game you will:

- Watch images of animals/furniture/vegetables.
- Rate tags that other players have given.
- Add new tags that you think are missing.

**Audio**. Rate & Tag emotions! Thanks for participating in this game! In this game you will:

- Listen to a speech fragment and focus on the emotional content of the recording.
- Rate tags that other players have given.
- Add new tags that you think are missing.

**Video**. Rate & Tag activities! In this game you will:

- Watch a video and focus on the activities happening.
- Rate tags that other players have given.
- Add new tags that you think are missing.

Participants then received further instructions regarding the rules of the game

**Images**. After watching the animal/furniture/vegetable you will see tags given by other players that describe the animal/furniture/vegetable. You should rate the relevance of each tag by clicking the appropriate amount of stars (1 star not very relevant, 5 stars very relevant). If you think that the tag is a mistake or completely irrelevant, you should flag it by clicking the flag icon. If you are the first person seeing this animal/furniture/vegetable, you may see no previous tags. You can also add your own tag that is relevant to describe the animal/furniture/vegetable. Your tag will then be rated by other players who are playing the game simultaneously.

**Audio**. After listening to the recording, you will see tags given by other players that describe the emotions in the speech fragment. You should rate the relevance of each tag by clicking the appropriate amount of stars (1 star not very relevant, 5 stars very relevant). If you think that the tag is a mistake or completely irrelevant, you should flag it by clicking the flag icon. If you are the first person listening to this speech sample, you may see no previous tags. You can also add your own tag that is relevant to describe the emotions in the speech fragment. Your tag will then be rated by other players who are playing the game simultaneously.

**Video**. After watching the video, you will see tags given by other players that describe the activities in the video. You should rate the relevance of each tag by clicking the appropriate amount of stars (1 star not very relevant, 5 stars very relevant). If you think that the tag is a mistake or completely irrelevant, you should flag it by clicking the flag icon. If you are the first person watching this video, you may see no previous tags. You can also add your own tag that is relevant to describe the activities in the video. Your tag will then be rated by other players who are playing the game simultaneously.

Finally, participants received the following guidelines regarding the tag input and the bonus scheme:

**Keep tags short.** A word like "green grass" should rather be submitted as "green" and "grass", whereas a compound word such as "red wine" cannot be separated, since "red wine" means something different than just "red" and "wine".

**Bonus rules.**

- If the tag you provide gets rated as a relevant tag (i.e., not flagged) by other players
- If your tag is unique and have not been introduced by others

| Modality | STEP | Captions |
|----------|------|----------|
| Audio | 230 | 187 |
| Video | 264 | 291 |

Table 3: Median of overall participants' time spent per stimulus (in seconds).

*Note:* Simply writing many and irrelevant tags is not a good idea because other players might flag your tag. Your experiment will terminate early if there are too many red flags!

Please try to use a variety of words to describe the animal / furniture / vegetable / emotion in the speech fragment / activities in the video, and use the entire star rating scale for your responses.

### B.6.1 VALIDATING STEP-TAG

We conducted a small, exploratory ablation study to validate STEP-Tag as a procedure for collecting diverse, accurate, and informative tags. First, we compared using multiple tags from the last iteration of STEP-Tag to using just a single randomly-selected highly-rated tag from the last iteration. We found that using a single tag greatly decreased correlation with human similarity (i.e., for the video dataset, the best-performing method on multiple tags had a correlation of $r = 0.74$ while the best-performing method on single labels had a correlation of $r = 0.35$). Second, we compared tags from the first iteration of STEP-Tag (equivalent to collecting tags without an adaptive procedure) to tags from the last iteration. We found that using first iteration tags greatly decreased correlation with human similarity (i.e., for the video dataset, the 'Tags CNNB mean (no split)' method, the correlation from the last iteration was $r = 0.74$ and from the first iteration it was $r = 0.44$; for 'Tags overlap' it was $r = 0.56$ from the last iteration and $r = 0.38$ from the first iteration). Finally, we extracted the Kinetics-200 labels for each video to compare the tags from STEP-Tag against the kinds of labels typically collected for machine learning datasets. We found that using labels decreased the correlation with human similarity (i.e., the best-performing method on pipeline tags had a correlation of $r = 0.74$ while the best-performing method on dataset labels had a correlation of $r = 0.64$).

### B.7 DURATION OF STEP-TAG AND CAPTIONS

To compare STEP-tag and captions, we computed the median of overall participants' time spent per stimulus (see Table 3). The times were only collected for the audio and video modality (captions for the image datasets were already collected by Marjieh et al. (2022)). We see that both methods consume roughly similar amounts of time, which is desirable as our analysis suggests that in some domains (e.g., video) tags yield the best results whereas in others (e.g., audio) captions do.

## C PREDICTION METHODS

We used two main types of methods to predict human similarity judgments. The first class ("DNN-based methods", described in section C.1) make use of pre-trained embedding models. In the second class of models ("Word Frequency Analysis methods", described in the section C.2) simple feature extraction techniques are used instead of pre-trained deep learning models. Figure 1 depicts schematic overview of all prediction methods that we used.

### C.1 DNN-BASED METHODS

The DNN-based methods use various embeddings and deep learning representations to predict human similarity judgments. These methods could be further split into three groups based on the kinds of input data they process, namely if they use a single sensory modality that is either image, audio or video ("unimodal models"; see subsection C.1.1), or use text that is either tag or captions ("text embeddings"; see subsection C.1.2), or use both ("multimodal models"). In addition, we also tested the performance of "stacked" representations, where the sensory and textual embedding of a select

number of models were concatenated into a single long embedding. Overall, the computation time of embedding methods took about two weeks on an x1.16xlarge Amazon Web Services instance with 64 vCPUs and 976 GiB of memory.

### C.1.1 Unimodal DNN-based methods

Table 4: All 30 image baseline models occurring in the top 50 best models reported in Figure 3A.

|  | Model name | Average score | SD score | Top 1 accuracy | Number of parameters (M) |
|---|---|---|---|---|---|
| 1 | Swin | 0.66 | 0.06 | 81.52 | 23.37 |
| 2 | ConvNeXT | 0.64 | 0.07 | N/A | 348.15 |
| 3 | NF-ResNet | 0.62 | 0.04 | 80.65 | 23.51 |
| 4 | NFNet l0 | 0.61 | 0.08 | 82.75 | 32.77 |
| 5 | ResNetV2 | 0.60 | 0.11 | N/A | 928.34 |
| 6 | NF-RegNet | 0.59 | 0.05 | 79.29 | 9.26 |
| 7 | VGG16 | 0.58 | 0.11 | 73.35 | 134.27 |
| 8 | VGG19 | 0.58 | 0.11 | 74.21 | 139.58 |
| 9 | ViT | 0.58 | 0.12 | 75.95 | 6.16 |
| 10 | ResMLP | 0.57 | 0.07 | 83.59 | 128.37 |
| 11 | Twins-SVT | 0.57 | 0.06 | 81.68 | 23.55 |
| 12 | Twins-PCPVT | 0.57 | 0.04 | 81.09 | 23.59 |
| 13 | VGG13 | 0.57 | 0.11 | 71.59 | 128.96 |
| 14 | CaiT | 0.57 | 0.04 | 82.19 | 17.18 |
| 15 | VGG11 | 0.57 | 0.10 | 70.36 | 128.77 |
| 16 | gMLP | 0.56 | 0.06 | 79.64 | 19.17 |
| 17 | PIT | 0.56 | 0.03 | 78.19 | 10.23 |
| 18 | DeiT | 0.56 | 0.03 | 72.17 | 5.52 |
| 19 | ConViT | 0.56 | 0.03 | 73.11 | 5.52 |
| 20 | TNT | 0.56 | 0.03 | 81.52 | 23.37 |
| 21 | CoaT | 0.55 | 0.04 | 78.43 | 5.35 |
| 22 | gMixer | 0.55 | 0.05 | 78.04 | 24.34 |
| 23 | XCiT | 0.55 | 0.04 | 82.57 | 11.92 |
| 24 | IG ResNeXt | 0.53 | 0.13 | 85.44 | 826.36 |
| 25 | Visformer | 0.52 | 0.02 | 82.11 | 39.45 |
| 26 | RepVGG | 0.52 | 0.11 | 80.21 | 81.26 |
| 27 | CLIP image | 0.50 | 0.11 | N/A | 102.01 |
| 28 | JXNesT | 0.50 | 0.07 | 81.42 | 16.67 |
| 29 | ECAResNet | 0.47 | 0.14 | 80.45 | 28.11 |
| 30 | DenseNet | 0.47 | 0.12 | 74.74 | 6.95 |

*Note.* Performance accuracy on ImageNet was based on Wightman (2019) and was not available for all models.

**Image models** We used 560 pre-trained models from the Pytorch Image Models (`timm`) repository (Wightman, 2019). We chose this repository as it contains an extensive and highly diverse set of pre-trained models in terms of architecture backbones, model sizes, and training sets. The repository includes models published from 2014 to 2022 that use various training sets (such as ImageNet1k, ImageNet21k, Instagram, etc.), training procedures objectives (e.g., pre-training, fine-tuning, self-supervision, weak supervision, etc.) and architectures (e.g., VGG, ResNet, Inception, Transformer, etc.). The repository also reports various evaluation metrics for each model (e.g., their ImageNet performance).

For each model, we computed the embedding from the last layer (typically before the final softmax layer; see below and Figure 10 for a preliminary analysis for the effect of layer depth in audio models). We then computed the cosine similarity between pairs of embedding vectors to produce a similarity matrix. The entire list of the performance of all models is detailed in the OSF repository

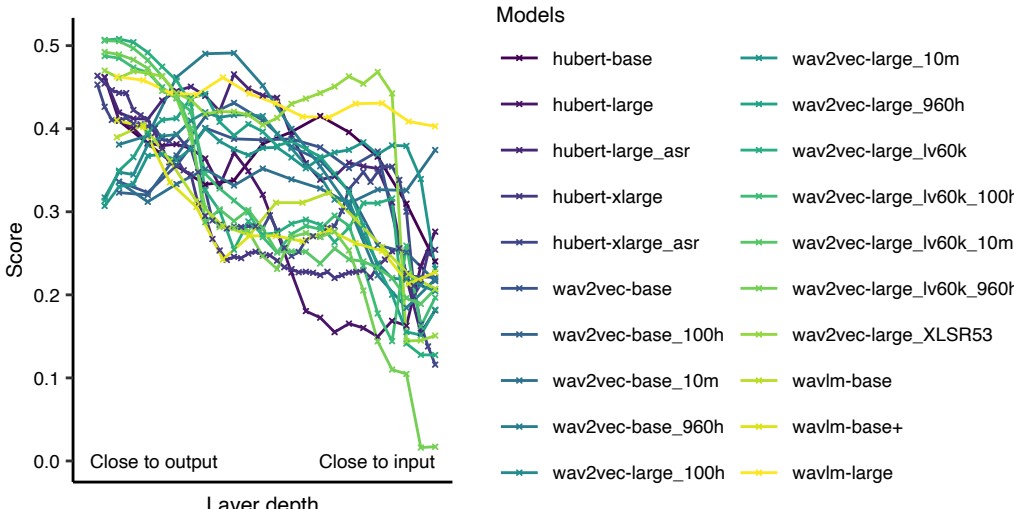

Figure 10: Scores for individual layers of audio models scaled to the total number of layers. Models are colored by their meta architecture.

associated with this paper [7]. Table 4 presents additional details for the top 42 image baseline models in Figure 3A including their average score (correlation to human judgments) across the three image datasets, the standard deviation (SD) of this score (across datasets, repeated runs and available model parameters in Wightman (2019)), their ImageNet accuracy, and their number of trainable parameters.

Figure 4A shows the correlation to human similarity as a function of the number of parameters for all 569 models. In general, we found that models that have more parameters perform better (Figure 4A). Plotting all the embedding technique correlations against the number of training parameters of their respective models showed statistically significant positive correlation ($r = 0.39, p < 0.001$). However, one possible explanation for this could be the improved performance of newer models, which typically have more parameters, on various computer vision tasks. To test this, we computed the performance (i.e., correlation with human similarity) of the various models as a function of their accuracy on ImageNet (Deng et al., 2009) - which was provided in Wightman (2019) for all models except for CLIP (whose implementation came from a different repository) as summarized in Figure 4B. We found a positive correlation between the two metrics ($r = 0.26, p < 0.001$), though with some clear exceptions. For example, the vision transformer BEiT (Bao et al., 2021) and the convolutional architecture EfficientNet (Tan & Le, 2019) achieved high accuracy on ImageNet but performed poorly on human data. On the other hand, the vision transformer Swin (Liu et al., 2021) and the convolutional architecture ConvNext (Liu et al., 2022) both performed well on ImageNet and human similarity. This suggests that architecture and number of parameters are better predictors of similarity judgments than performance on ImageNet. Further analysis is required to determine what kind of architectural components actually contribute to more human-like performance (Langlois et al., 2021).

**Audio models** We used all pre-trained wav2vec 2.0 (Baevski et al., 2020) and HuBERT (Hsu et al., 2021) models available in `torchaudio` (Yang et al., 2021). We also extracted embeddings from WavLM (Chen et al., 2021) and data2vec audio models (Baevski et al., 2022). Furthermore, we used additional wav2vec 2.0 and HuBERT models that were either specialized on emotion recognition or speaker identification (wen Yang et al., 2021; Wagner et al., 2022; Ravanelli et al., 2021). The performance of HuBERT, wav2vec 2.0, and WavLM models is shown in Figure 3B. Additional details about the models are displayed in Table 5.

In addition, we explored the correlation between the audio models and human similarity data as a function of the layer in the model. Earlier literature has suggested that similarity to human representations may depend on the layer of the model (Kell et al., 2018; Yamins et al., 2014; Yamins,

---

[7]https://osf.io/kzbr5/?view_only=3dea58e008ce41c290ef0f374bdbf444

Table 5: All audio baseline models used in the analysis.

| | Model name | Emotion correlation | Number of parameters (M) |
|---|---|---|---|
| 1 | wav2vec 2.0 lv60k (100h) | 0.49 | 317 |
| 2 | wav2vec 2.0 lv60k (960h) | 0.49 | 317 |
| 3 | wav2vec 2.0 lv60k | 0.51 | 317 |
| 4 | wav2vec 2.0 lv60k (10m) | 0.51 | 317 |
| 5 | HuBERT xlarge ASR | 0.45 | 1000 |
| 6 | HuBERT xlarge | 0.46 | 1000 |
| 7 | HuBERT large ASR | 0.46 | 300 |
| 8 | wav2vec 2.0 large XLSR53 | 0.47 | 317 |
| 9 | HuBERT large | 0.46 | 300 |
| 10 | wav2vec 2.0 (Audeering, emotion) | 0.49 | 317 |
| 11 | HuBERT base | 0.41 | 90 |
| 12 | WavLM large | 0.46 | 316.62 |
| 13 | HuBERT base (superb, emotion) | 0.42 | 90 |
| 14 | HuBERT base (superb, speaker) | 0.42 | 90 |
| 15 | WavLM base+ | 0.41 | 94.70 |
| 16 | wav2vec 2.0 base (960h) | 0.38 | 95 |
| 17 | WavLM base | 0.39 | 94.70 |
| 18 | wav2vec 2.0 base | 0.34 | 95 |
| 19 | wav2vec 2.0 base (10m) | 0.34 | 95 |
| 20 | wav2vec 2.0 base (superb, emotion) | 0.34 | 95 |
| 21 | wav2vec 2.0 base (superb, speaker) | 0.34 | 95 |
| 22 | wav2vec 2.0 base (100h) | 0.32 | 95 |
| 23 | HuBERT large (superb, emotion) | 0.29 | 300 |
| 24 | HuBERT large (superb, speaker) | 0.29 | 300 |
| 25 | wav2vec 2.0 large (100h) | 0.32 | 317 |
| 26 | wav2vec 2.0 large (superb, emotion) | 0.31 | 317 |
| 27 | wav2vec 2.0 large (superb, speaker) | 0.31 | 317 |
| 28 | wav2vec 2.0 large (960h) | 0.31 | 317 |
| 29 | wav2vec 2.0 large (10m) | 0.31 | 317 |
| 30 | data2vec audio large (960h) | 0.31 | 313.28 |
| 31 | data2vec audio base (100h) | 0.23 | 313.28 |
| 32 | data2vec audio large (100h) | 0.23 | 313.28 |
| 33 | data2vec audio large (10m) | 0.21 | 313.28 |
| 34 | wav2vec 2.0 (SpeechBrain, emotion) | 0.11 | 95 |
| 35 | data2vec audio base (960h) | 0.16 | 93.16 |
| 36 | data2vec audio base (10m) | 0.15 | 93.16 |

2020). We expected that the layers closer to the input of the model (where the representation is more low-level) to be less predictive. In general, we found that this was the case (Figure 10). In some variants of wav2vec, however, intermediate representations performed better, possibly due to the misalignment of the training task of wav2vec with the emotion task. This analysis confirms the choice we made in the paper to mostly use the last two layers of the models. Preliminary analysis of the image and video models also explored different layers, but the results were similar to those we presented in audio, and are therefore not reported here.

**Video models** We extracted embeddings from the 'Slow' (a 3D ResNet; see Feichtenhofer et al. (2019)), Slowfast (a 2-path model with one path capturing semantics and the other capturing fine details; see Feichtenhofer et al. (2019)), and X3d (a model that initially start as a simple 2D image classifier but is expanded in several axes; see Feichtenhofer (2020)) architectures implemented in `pytorchvideo` (Fan et al., 2021). All video models were pre-trained on the Kinetics-400 dataset (Kay et al., 2017). The performance of the models is displayed in Figure 3C. Numeric correlation values are detailed in Table 6 along with model accuracy (Top1 and Top5) on Kinetics-400, and the number of parameters in each model. The accuracies and parameter counts are listed as reported in

Fan et al. (2021). As with previous modalities, the number of parameters appears to be positively correlated with correlation to human similarity.

Table 6: All video baseline models used in the analysis.

|   | Model name | Correlation | Kinetics-400 Top1 Acc | Kinetics-400 Top5 Acc | Number of parameters (M) |
|---|---|---|---|---|---|
| 1 | Slowfast r50 | 0.65 | 76.94 | 92.69 | 34.57 |
| 2 | Slowfast r101 | 0.64 | 77.90 | 93.27 | 62.83 |
| 3 | Slow r50 | 0.61 | 74.58 | 91.63 | 32.45 |
| 4 | X3d M | 0.53 | 75.94 | 92.72 | 3.79 |
| 5 | X3d S | 0.49 | 73.33 | 91.27 | 3.79 |
| 6 | X3d XS | 0.48 | 69.12 | 88.63 | 3.79 |

### C.1.2 TEXT EMBEDDING METHODS

**Caption text embedding.** Since there are multiple captions per stimulus, an aggregation procedure had to be applied to produce a single embedding vector for each stimulus. In our main analysis, for each stimulus, we extracted the embedding for each associated caption and averaged these embeddings together before computing cosine similarity between the mean embeddings. We also tried an alternative approach of concatenating the captions together into a single paragraph, which we then passed through the LLMs to compute a single embedding per stimulus. We found that this did not consistently improve performance and in many cases even decreased it, though we note that we did not experiment with different permutations of the concatenated captions, nor did we extensively study other ways to combine them together. Future work could explore other techniques for pre-processing captions and aggregating representations from multiple captions in ways that would improve correlation with human similarity judgments.

**Tag text embedding.** We experimented with several algorithms for computing similarity between sets (or multi-sets) of tags. The algorithms described in this section all involve using ConceptNet NumberBatch (CNNB) (Speer et al., 2017) as the embedding backbone for turning discrete tags into continuous vector representations. For each stimulus, we took the tags remaining in the final iteration, and tested whether they were found in the dictionary for our embedding model. If a tag was not found and if it contained no spaces, we tried to correct the spelling before trying to look it up in the dictionary again. If a tag contained spaces, we split it into individual words, correct their spelling, and averaged together the embedded representations of those words that were found in the dictionary. Tags that were not found even after spelling correction and splitting were excluded from the set and did not contribute to the final representation. For the methods marked '(no split)' we did not split multi-word tags, instead we just excluded multi-word tags that were not found in the embedding model dictionary. In the following, we describe the different techniques used to generate predictions based on tag embeddings.

*Tags CNNB overlap.* For each pair of stimuli, we counted the number of 'almost identical' tag embeddings, defined as every respective element of the two embeddings being less than a certain threshold apart (in our case, this threshold was 0.1). We then set similarity for that pair of stimuli to be this count, i.e., the number of 'almost identical' tags, normalized by the total number of tags across the respective two sets.

*Tags CNNB quantized.* This method involves quantizing tags using cosine similarity to find the number of unique tags. For each pair of stimuli, we counted the number of tags assigned to the first stimulus that had cosine similarity greater than a certain threshold (in our case, this threshold was 0.7) to at least one tag of the second stimulus (call this value $N_A$) and vice-versa ($N_B$). The minimum of these two values is the number of unique, shared tags between the two sets ($\min(N_A, N_B)$). The total number of unique tags across the two sets is then the total number of tags in each set ($T_A + T_B$) minus the maximum number of shared tags ($\max(N_A, N_B)$). We compute similarity as the ratio of the number of unique, shared tags to the total number of unique tags, $S_{AB} = \frac{\min(N_A, N_B)}{T_A + T_B - \max(N_A, N_B)}$. For example, suppose the two sets of tags are $A : \{a, b, c, g\}$ and $B : \{a, b, d, e\}$, so $T_A = T_B = 4$, and that $a, c$ have cosine similarity of 0.8. The number of tags from set A found in set B is $N_A = 3$, and those from B found in A is $N_B = 2$. The number of unique, shared tags is $\min(N_A, N_B) = 2$ (since

$\{a, b, c\}$ can be represented by $\{a, b\}$), and the total number of unique tags is $4 + 4 - 3 = 5$ (since $\{a, b, c, g, a, b, d, e\}$ can be represented by $\{a, b, d, e, g\}$). The assigned similarity is then $S_{AB} = \frac{2}{5}$.

*Tags CNNB mean.* The set of tag embeddings for each stimulus were averaged together to form a single embedding assigned to the respective stimulus. We then computed cosine similarity on the embeddings of each pair of stimuli.

*Tags CNNB mean (no split).* Same as above, but without splitting multi-word tags (i.e., ones that contain spaces) during the embedding process.

All spelling corrections in the algorithms listed above were performed using the Python package `pyspellchecker`[8], taking the top corrected recommendation returned by the spell checker in each case.

*Tags to caption Roberta (SimCSE).* Additionally, for the images datasets, we experimented with converting sets of tags into captions and then using those captions with our best-performing LLM ('sup-simcse-roberta-large') the same way we do with user-generated captions. To convert a set of tags into a caption, we joined the set of tags with commas and prepended them with the phrase "This is an image of".

### C.1.3 OTHER DNN-BASED METHODS

For the image datasets, we also considered several other methods that made use of DNNs but do not fit into the categories described above.

**GPT3 prompting** We experimented with prompting GPT3 (Brown et al., 2020), a large pre-trained language model, to directly output similarity judgments as a text-completion problem rather than having to access model embeddings as we did above. We used a few-shot prompting approach where in each prompt we included three context examples of pairs of tag sets and their associated similarity rating. We then provided the pair of tag sets for the two images that we wanted to get a similarity rating for but left the rating empty for the model to fill in.

Here is an example prompt with the GPT3 response bolded and in square brackets:

---

People described pairs of images using words.
How similar are the two images in each pair on a scale of 0-1 where 0 is completely dissimilar and 1 is completely similar?

Here are the descriptions of image one: tortoise, slow, protected, shell, turtle, scaly, old, cold-blooded
Here are the descriptions of image two: monkey, ape, mammal, black and white, hairy, agile, primate, smart, tree-dwelling
Rating: 0.05

Here are the descriptions of image one: rhinoceros, horn, gray, standing, heavy body, endangered, wild, africa, african
Here are the descriptions of image two: tiger, open mouth, stripes, feline, predator
Rating: 0.27

Here are the descriptions of image one: goat, eye, leg
Here are the descriptions of image two: mammal, wide-nosed, mandrill, primate, baboon, smart
Rating: 0.19

Here are the descriptions of image one: black, primate, mammal, hairy, chimpanzee, africa, african, great ape, smart, omnivore
Here are the descriptions of image two: zebra, striped, two-toned, wild, staring, mammal, equine, herd animal, africa
Rating: **[0.14]**

---

We repeated this four times for each pair of images in each image dataset with a different set of context examples during each repetition and averaged together the GPT responses to get a final

---

[8]`https://pyspellchecker.readthedocs.io/en/latest/`

similarity prediction for each pair. In total, creating the context examples required having access to human similarity judgments over only 12 pairs of images. We found that this approach yielded surprisingly good predictions, with an average correlation of $r = 0.62$ across the image datasets. We believe this approach merits future investigation to determine whether prompt engineering can further increase the performance.

**Image captioning models**    We experimented with using pre-trained image captioning models to generate captions for our images and then using those captions with our best-performing LLM ('sup-simcse-roberta-large') the same way we do with user-generated captions. We used three pre-trained image captioning models from HuggingFace ('flamingo-mini', 'vilt-b32-finetuned-vqa', and 'vit-gpt2-image-captioning') to generate text descriptions for our images. However, the performance was quite poor with an average of $r = 0.29$ across the three models. As a result, $O(N)$ language-based methods cannot easily be reduced to $O(1)$ even when domain-relevant pre-trained caption models are available.

## C.2    WORD FREQUENCY ANALYSIS METHODS

In this work, we also conducted an additional evaluation of prediction models beyond embedding-based techniques (described in the previous section). Specifically, we compared the predictions of embedding-based models, which utilize deep learning representations, with those of traditional methods of text mining.

Before the word frequency analysis, we performed the following initial pre-processing steps

- For caption data, we concatenated all the captions describing the same stimulus into a single long "document."

- For tag data, we wanted to prioritize tags that appeared earlier in the tag-mining chains and were rated higher. To that end, we gathered all tags from all iterations and duplicated tags from a given iteration based on the ratings they received. For example, if the tag "tomato" received three stars, then we would add the repeated tokens "tomato, tomato, tomato" to the aggregated list ("document"). In a given iteration, flagged tags are removed, but if they are rated later, then they are included. The total number of repetitions per token is equal to the sum of all the stars they received in all iterations. As a result, each token is repeated multiple times, which we take into consideration in consequent analysis.

For the next steps, we used the Matlab text analytics toolbox [9]. Unless otherwise specified, we used default parameters for all functions. To generate similarity matrices, we applied the following methods:

**Co-occurrence method**. In this approach, we simply counted the number of repeated pairs of words in documents $i$ and $j$ and normalized by the total number of pairs. Formally, we use $w_i$ to denote the word list of a document $i$. Let $w_{i,k}$ be the $k$-th word in the $w_i$ list of words, and let $|w_i|$ denote the length of the list. We denote by $\delta(c, d)$ the indicator function that returns 1 if and only if the word $c$ is identical to the word $d$, and 0 otherwise. We computed the co-occurrence score $S(w_i, w_j)$ according to the following formula:

$$S(w_i, w_j) = \frac{\sum_k \sum_l \delta(w_{i,k}, w_{j,l})}{|w_i||w_j|}$$

We suggest using this method only with tags and not with captions.

**Co-occurrence-rep**. This method was applied only to tags. We used an identical procedure to the *Co-occurrence* method, except that we did not separate the words within a tag as separate tokens and instead treated the entire tag (that may include multiple words) as a single token.

**Rouge score**. In this approach, similarity was estimated by computing the rouge score of the word lists associated with each pair of documents. The Rouge score was computed using `rougeEvaluationScore` (Rouge, 2004). We suggest using this method only with tags and not with captions.

---

[9]`https://mathworks.com/products/text-analytics.html`

The following methods make use of tokenized data and a pre-processing procedure that we found effective. Pre-processing was applied to both tag and caption data and tokenization was performed as follows:

- We separate all text into single words by applying the `tokenizedDocument` function.
- We added part of speech information using the `addPartOfSpeechDetails` function.
- We performed Lemmatization using the `normalizeWords` function.
- We erased punctuation from the token using the `erasePunctuation` function.
- We removed stopwords using the `removeStopWords` function.
- We removed words with less than two characters or more than 15 characters.
- We created a bag of words representation of each tokenized document using the `bagOfWords` function.
- We also removed words that were not present in more than two documents using the `InfrequentWords` function.

With the results of these pre-processing steps, we then computed similarity matrices based on the following methods:

**bm25S**. We used `bm25+` to compute similarity between documents (Barrios et al., 2016) using Matlab's `bm25Similarity` function. This function represents TF-IDF-like retrieval functions used in document retrieval. We used a variant that has a normalization function that properly handles documents with a long list of words.

**tf-idf-cosine**. We computed pairwise cosine similarities between document pairs using the TF-IDF matrix derived from their word counts and Matlab's `cosineSimilarity` function.

## C.3 SUPERVISED METHODS

Several previous studies investigated improving correlations by applying and fine-tuning simple linear transformations to embedding vectors $z^T \mathbf{W} z$ where $\mathbf{W} = \text{diag}(w_1, \ldots, w_d)$ via a cross-validated ridge regression procedure that could be fit to ground-truth similarity judgments. The parameters of the diagonal reweighting matrix $\mathbf{W}$ are fitted to a training subset of stimuli and used to predict similarity of pairs in a held-out validation set Peterson et al. (2018); Marjieh et al. (2022). To be consistent and make results comparable, here we report the results of performing this 6-fold cross-validated linear transformation (LT-CCV) on the model embeddings and datasets considered in this work. The analysis was carried out using the `RidgeCV` package from the `scikit-learn` Python library Pedregosa et al. (2011). Results with both normalized ('LT CCV (norm)') and un-normalized ('LT CCV') regressors are shown in Figure 11; see `RidgeCV` documentation for details on normalization [10]. We see that the linear transformation does not consistently improve performance (and can even decrease it) when applied to many of the modality-based or stacked embeddings, but it does frequently improve performance when applied to caption embeddings. Due to their instability and risk of overfitting, we do not use these methods in our main analysis.

---

[10]`https://scikit-learn.org/stable/modules/generated/sklearn.linear_model.RidgeCV.html`

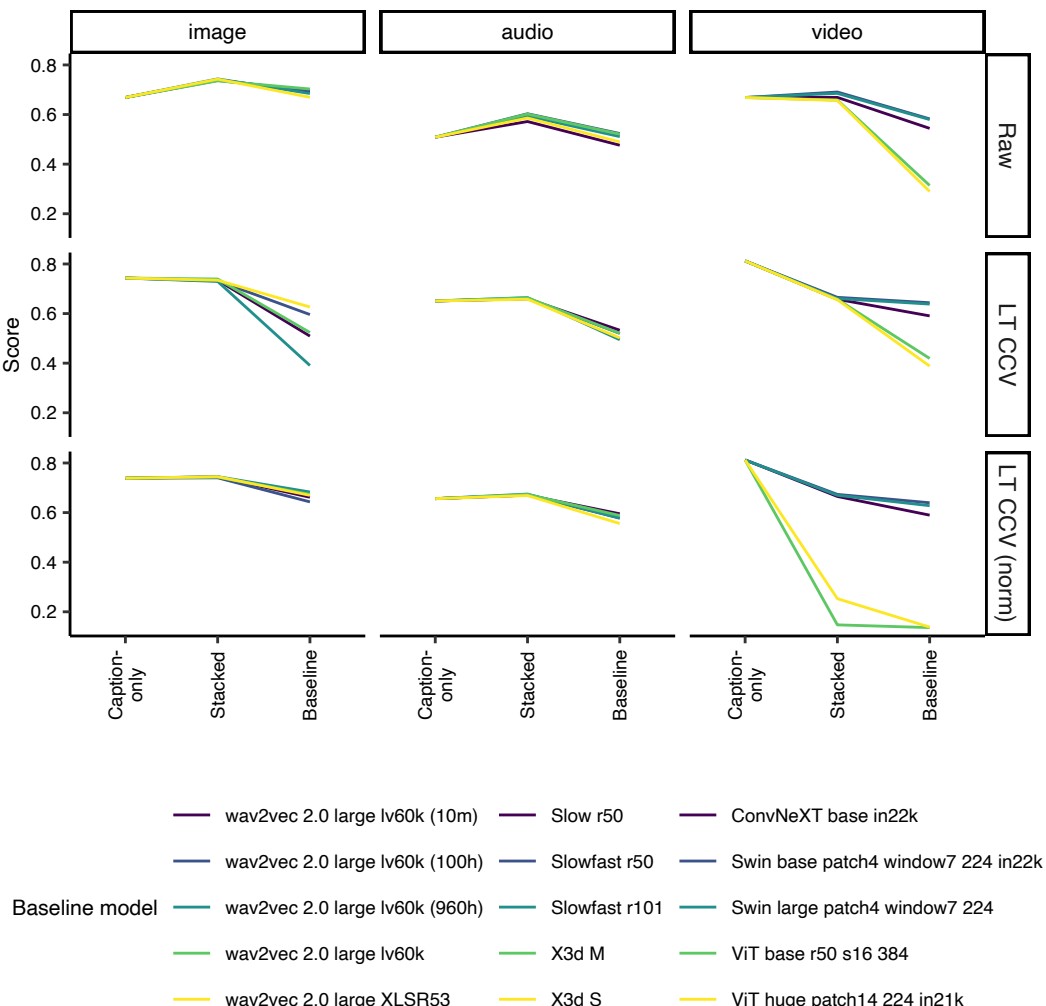

Figure 11: Effect of fine-tuning model embeddings using a small subset of similarity judgments.

