# OpenReview forum: "Words are all you need? Language as an approximation for human similarity judgments"
_ICLR.cc/2023/Conference — ICLR 2023 poster_

### Official Review · Reviewer_AJDz · 2022-10-13

**Confidence:** 2
**Correctness:** 3
**Technical Novelty And Significance:** 2
**Empirical Novelty And Significance:** 3
**Recommendation:** 8

**Clarity, Quality, Novelty And Reproducibility:**

As I mentioned, the novelty is somewhat limited because of the prior human
judgements work the authors get their data from. But, the extensive set of
experiments is interesting/new.

**Strength And Weaknesses:**

Reconstructing pairwise human similarity judgments across N objects is
an interesting task, and the author's idea of gathering textual
descriptions is clever. The performance of the stacked models (which
depend on domain-specific pretrained models and text-text
similarities) and text-only models are surprisingly strong across a
variety of domains --- this finding has implications for the relative
surprising representational strength of text. The annotation process
they describe for their tag corpus is interesting --- multiple rounds
of annotations are undertaken with crowdworkers fixing each-other's
errors. The authors consider a large number of models across many
domains --- the benchmark experiments are extensive, and the "size
vs. accuracy" plots in the end are cool. Figure 5 is amazing --- few
works offer such clear practical guidance.  Finally, I liked the
hopeful message in the end --- that a combined multimodal approach
seems to work best, so don't completely discard the domain-specific
representations.

It should be noted that the cited work where some of the data is from,
Marjieh et al. 2022, is quite similar in the sense that it also
proposes to estimate human similarity judgments. This work appears to
extend that work by exploring more types of models and their
combinations.

My main concerns are that this set of experiments suggests some clear
next setups that I don't nesc. think are out of scope for this work.
Specifically:

- If the goal in practice is to reconstruct human judgments --- I
  would have liked the authors to compare against a supervised setup.
  How much of the remaining misalignments can be fixed by gathering n
  << N^2 additional pairwise judgments and then training a supervised
  model ontop of the models?

- I would have liked to have seen more experiments with even less
  supervision --- is it possible to gather n < N captions, and train a
  model on those very few pairs to map to text, and then use the
  resulting modality --> text model as input to the models shown here?

- I would have appreciated more discussion of tags vs. captions for
  the images. In figure 3A, I assume that the text only methods use
  captions unless specified otherwise. But --- because the tags were
  not fed to, e.g., BERT/RoBERTa, I can't tell if the performance
  gains are due to the full captions being used, or if the gains are
  due to the more expressive models. A trivial linearization of, e.g.,
  "A photo of a tag1, tag2, and tag3" handed to a LLM would have been
  a nice comparison to see.

- Finally, I would have been interested in a fully LLM setup where
  instead of pairwise cosine similarities, both text pairs are fed to
  a very large LLM (e.g., OPT or GPT3) and the model outputs a likert
  rating as text.

**Summary Of The Paper:**

The authors consider the task of approximating human didactic
similarity judgments over N pairs of (images/audio/text). While high
quality human judgments over all pairs is ideal, (N choose 2) is too
big for large N. The authors consider gathering N descriptions of the
objects (tags or captions), extracting text representations of those
descriptions, and then computing text-text similarity. Experiments
across a large number of models and datasets reveal the surprising
effectiveness of computing text-text similarities.

**Summary Of The Review:**

Overall, this work address an interesting task/approach ---
approximating pairwise similarity judgments over N objects with only
O(N) annotation via language models. While this setup has been
considered by Marjieh, this work presents more experiments that show
the text representations are surprisingly strong. They consider a very
large number of models, modalities, etc. The released tag sets would
likely be useful to someone. But, a handful of missing additional
experiments, which are arguably in-scope for the present work, are not
presented. And, prior work has already made the observation that LLMs
can approximate similarity judgments across non-textual modalities,
which appears to somewhat limit the novelty of this work.


After response:

The authors seemingly ran all of the experiments I suggested and have incorporated them into the paper. Amazing :-)
Furthermore, the authors clarified the difference with prior work.
All my concerns are addressed, so I raised my score from 5-->8

---

> ### Author Response · Authors · 2022-11-18
> **Response to Reviewer AJDz**
>
> We would like to thank the reviewer for their detailed evaluation, and especially for proposing follow-up experiments that we believe have substantially enhanced our work. We detail those additional experiments as well as responses to other points raised by the reviewer in what follows.
>
> ### Reviewer AJDz - Difference from Marjieh et al. 2022
> *“It should be noted that the cited work where some of the data is from, Marjieh et al. 2022, is quite similar in the sense that it also proposes to estimate human similarity judgments. This work appears to extend that work by exploring more types of models and their combinations.”*
>
> **Response**: our work differs from Marjieh et al. (2022) in a few significant ways which we highlight below:
> 1. **Methodological innovation**: we proposed STEP-Tag, an adaptive paradigm for domain-general tag mining which provides users with a method to collect high-quality annotations. Marjieh et al. (2022) relied on the availability of such annotations.
> 2. **Comprehensive baseline evaluation**: we evaluated 600+ baseline DNN models across three different modalities against multiple language models. Marjieh et al. (2022) considered one modality, a single language model, a single word embedding model and one baseline (CNN).
> 3. **Stacking**: we introduced stacked representations and showed that they predict human judgments best. This aspect is completely absent from Marjieh et al. (2022).
> 4. **Embedding-free methods**: we evaluated word-frequency-based methods and showed that they can provide good proxies for human judgments. Crucially, combined with STEP-tag these provide a toolkit for predicting human similarity in low-resource languages. This was not tested by Marjieh et al. (2022).
> 5. **Released dataset**: we provide a publicly available dataset containing more than 200,000 human judgments and all model predictions that can be used by other researchers.
> 6. **Guide**: we synthesized our findings into an explicit guide that provides an easy way to construct an action plan for researchers interested in approximating human similarity judgments.
> 7. **Additional text-methods proposed by the reviewer**: these include prompt-based predictions with GPT3 and O(1) text-only predictions with image-to-text captioning models (see below).
> To summarize, while Marjieh et al. (2022) perform an initial small scale exploration of the question of predicting similarity based on text, our work substantially expands this idea into a polished, extensively tested and streamlined process that can be directly used by ML practitioners and cognitive scientists in both low and high resource languages.
>
> ### Reviewer AJDz - Follow-up experiments
>
> We implemented all the reviewer’s suggestions for follow-up experiments and integrated them in the manuscript.
>
> *“If the goal in practice is to reconstruct human judgments --- I would have liked the authors to compare against a supervised setup. How much of the remaining misalignments can be fixed by gathering n << N^2 additional pairwise judgments and then training a supervised model ontop of the models?”*
>
> **Response**: We have revised the supplement to include such an analysis (Figure 11 and Section C.3). Specifically, we applied the supervised method proposed by Peterson et al. (2018) for improving alignment through a simple linear transformation. This is done by fine-tuning a linear transformation $\mathbf{W}=\textrm{diag}(w_1,\dots,w_d)$ that is applied to the embedding vectors $z$ so that predictions are generated via $z^T\mathbf{W}z$. The set of weights $\mathbf{W}$ are fitted on a small training subset of similarity judgments and then evaluated on a held-out validation set (we perform 6-fold cross-validation).  The results are summarized in Figure 11. We found that this indeed boosted the performance of some of our models, especially the caption-based ones, however, in certain cases (e.g., video) this appeared to have a negative effect on the performance of some of the stacked representations.
>
> **Response continued below.**

---

> > ### Author Response · Authors · 2022-11-18
> > **Response to Reviewer AJDz (Part 2)**
> >
> > ### Reviewer AJDz - Follow-up experiments (cont.)
> >
> > *“I would have liked to have seen more experiments with even less supervision --- is it possible to gather n < N captions, and train a model on those very few pairs to map to text, and then use the resulting modality --> text model as input to the models shown here?”*
> >
> > **Response**: This is a great idea since even O(N) can be fairly expensive when collecting a large enough dataset!  Based on the reviewer’s suggestion we now added this analysis to the paper (see C.1.3). Training a captioning model normally requires a large number of captions so for this experiment we used three pre-trained image captioning models from HuggingFace (flamingo-mini, vilt-b32-finetuned-vqa, and vit-gpt2-image-captioning) to generate text descriptions for our images and then fed those descriptions into our best-performing language model, SimCSE-Roberta, to generate predictions. This can be thought of as a ‘best case’ scenario since the captioning models we used were pre-trained on large captioning datasets (i.e. models trained on a much smaller number of captions n would at best perform as well as these models). Unfortunately, we found that the performance of these image-to-caption-to-similarity methods (average of r=0.29 on images) was far below the language-based O(N) methods, and even below many of the O(1) vision-based methods. We added these details to Sections 3.2 and C.1.3.
> >
> > *“I would have appreciated more discussion of tags vs. captions for the images. In figure 3A, I assume that the text only methods use captions unless specified otherwise. But --- because the tags were not fed to, e.g., BERT/RoBERTa, I can't tell if the performance gains are due to the full captions being used, or if the gains are due to the more expressive models. A trivial linearization of, e.g., "A photo of a tag1, tag2, and tag3" handed to a LLM would have been a nice comparison to see.”*
> >
> > **Response**: This is a good point. Our embedding-free word-frequency methods serve as a control for this as we apply the same procedure to tags and captions. In Figure 3A, you can see that captions with word-frequency methods perform better than tags with the same word-frequency methods. We also conducted the proposed tags-to-captions linearization and have added the results (r=0.55 which is comparable to using captions but not quite as good) to Figure 3A. Specifically, we converted tags into captions using the procedure described by the reviewer and then embedded them with our best-performing LLM (‘sup-simcse-roberta-large’) the same way we do with user-generated captions. We added the details to Sections 3.2 and C.1.2.
> >
> > *“Finally, I would have been interested in a fully LLM setup where instead of pairwise cosine similarities, both text pairs are fed to a very large LLM (e.g., OPT or GPT3) and the model outputs a likert rating as text.”*
> >
> > **Response**: Again, a great suggestion! We implemented this using GPT3 to generate such predictions using a few-shot prompting approach. Specifically, within the prompt, we included three context examples of pairs of tag sets and their associated similarity rating. We then provided the pair of tag sets for the two images that we wanted to get a similarity rating for but left the rating empty for the model to fill in.
> >
> > Here is an example prompt (model output is in square brackets):
> > ```
> > People described pairs of images using words.
> > How similar are the two images in each pair on a scale of 0-1 where 0 is completely dissimilar and 1 is completely similar?
> >
> > Here are the descriptions of image one: tortoise, slow, protected, shell, turtle, scaly, old, cold-blooded
> > Here are the descriptions of image two: monkey, ape, mammal, black and white, hairy, agile, primate, smart, tree-dwelling
> > Rating: 0.05
> >
> > Here are the descriptions of image one: rhinoceros, horn, gray, standing, heavy body, endangered, wild, africa, african
> > Here are the descriptions of image two: tiger, open mouth, stripes, feline, predator
> > Rating: 0.27
> >
> > Here are the descriptions of image one: goat, eye, leg
> > Here are the descriptions of image two: mammal, wide-nosed, mandrill, primate, baboon, smart
> > Rating: 0.19
> >
> > Here are the descriptions of image one: black, primate, mammal, hairy, chimpanzee, africa, african, great ape, smart, omnivore
> > Here are the descriptions of image two: zebra, striped, two-toned, wild, staring, mammal, equine, herd animal, africa
> > Rating: [0.14]
> > ```
> > We repeated this four times for each pair of images with a different set of context examples during each repetition and averaged together GPT’s responses to get a final similarity prediction for each pair. In total, creating the context examples required having access to human similarity judgments over only 12 pairs of images. We found that this approach yielded surprisingly good predictions, with an average correlation of r=.62 across the image datasets. We now incorporate this in Figure 3 and Section 3.2, and describe the details in Section C.1.3.

---

> > > ### Comment · Reviewer_AJDz · 2022-11-29
> > > **Thanks for the updates!**
> > >
> > > Hi There,
> > >
> > > Thanks for the updates! It's pretty cool that you were able to run *all* (!!) of the experiments I suggested --- you all work quite hard! :-)
> > >
> > > Thanks also for making clear the difference between this and prior work.
> > >
> > > I will update my score accordingly.

---

### Official Review · Reviewer_kd7y · 2022-10-24

**Confidence:** 4
**Correctness:** 4
**Technical Novelty And Significance:** 3
**Empirical Novelty And Significance:** 3
**Recommendation:** 8

**Clarity, Quality, Novelty And Reproducibility:**

The paper presents a straightforward but well-motivated idea. Having representations match human similarity judgement is indeed useful.

Implementing this technique would require (i) ability to acquire free-text or tag values which are domain-dependent but not prohibitively expensive, (ii) ability to obtain LLM representations which are straightforward since high quality implementations and libraries exist for these now. Reproducibility is not an issue.



**Strength And Weaknesses:**

Strengths: I think this is a well-motivated problem. Learned representations are often used as features in the small-data regimen or sometimes directly for getting proximity scores in an AI setting. This paper address the human interpretability of these representations by (i) confirming that human similarities and proximity scores from models can vary a lot, (ii) text-descriptions or tags can be leveraged and stacking these representations with the model-learned representations can help. I also appreciate that the technique is scalable and in many cases not that much of an overhead to implement. I appreciate the arguments in the related text that leverage cognitive science literature. In addition, the paper is easy to follow.

Weaknesses: The paper doesn't have too many weaknesses. I was wondering if we could get some numbers on if the stacked representations help in additional downstream tasks like say classification (i.e. does the performance on imagenet improve if you use imagenet + text). However, I understand that this can be significant undertaking and do not want to base my review on this experiment but it is a potential future direction.

**Summary Of The Paper:**

The authors make the observation that representations learned by DL models produce proximity scores very different from human evaluations. They introduce a simple and scalable technique to make the human and model produced similarity scores closer. Essentially, text descriptions or tags for various input data points (across modalities) are passed through LLMs (or word-frequency methods) and the resulting representations used for proximity scores. Stacked representations combining the existing model and the text description representation is shown to consistently match human similarity metrics better.

**Summary Of The Review:**

- Well motivated problem
- Clearly described technique that is scalable, easy to implement
- Techniques like these that are easy to implement and help with interpretability are of great use in the small-data regimen (where the bulk of us are). I would like to see this paper at ICLR.

---

> ### Author Response · Authors · 2022-11-18
> **Response to Reviewer kd7y**
>
> We are really thankful for the reviewer’s thorough and positive evaluation of our paper! We are grateful that the reviewer appreciates both the scalability of our methods and their applicability to small-data settings.
>
> ### Reviewer kd7y - Stacked representations and other downstream tasks
> *“I was wondering if we could get some numbers on if the stacked representations help in additional downstream tasks like say classification (i.e. does the performance on imagenet improve if you use imagenet + text). However, I understand that this can be significant undertaking and do not want to base my review on this experiment but it is a potential future direction.”*
>
> **Response**: We completely agree with the reviewer that this would be an interesting avenue to explore next. There is already some evidence from classification literature that multimodal classifiers (e.g. image and text) outperform unimodal ones (https://arxiv.org/abs/1909.02950, https://arxiv.org/abs/2112.03562, https://arxiv.org/abs/2205.01917 ). We hope to explore this direction further in the near future.

---

### Official Review · Reviewer_hUBS · 2022-10-24

**Confidence:** 3
**Correctness:** 3
**Technical Novelty And Significance:** 2
**Empirical Novelty And Significance:** 2
**Recommendation:** 5

**Clarity, Quality, Novelty And Reproducibility:**

The paper is generally well-written. The authors consider multiple modalities and datasets to evaluate their proposed method. But the novelty of the proposed method is weak.

**Strength And Weaknesses:**

Strength:
1. The paper is overall well-written and easy to follow.
2. Multiple datasets are evaluated for the proposed method.
3. Multiple modalities are explored and evaluated on the proposed method.

Weakness:
1. The technical contribution of the proposed method is weak.
2. No baselines are compared to the proposed method in the experiment section.

**Summary Of The Paper:**

The paper proposes a new class of similarity approximation methods based on language. To collect the language data required by these new methods, the authors also developed and validated a new adaptive tag collection pipeline, which is significantly cheaper than the classical methods. Finally, the authors also develop ‘stacked’ methods that combine language embeddings with DNN embeddings, and find that these consistently provide the best approximations for human similarity across all the three modalities.

**Summary Of The Review:**

The main issue of the paper is the novelty of the proposed method. Based on technical novelty and insufficient experiments, I don’t think the current version meets the standard of the ICLR.

---

> ### Author Response · Authors · 2022-11-18
> **Response to Reviewer hUBS**
>
> We thank the reviewer for their careful evaluation of our work and are glad they found it well-written and easy-to-follow. We detail our responses to the points raised below.
>
> ### Reviewer hUBS - Clarifying contributions
> *“The technical contribution of the proposed method is weak.”*
>
> **Response**: We want to emphasize the technical contributions of this paper:
> 1. Providing a simple and accessible approach for predicting human similarity judgment using O(N) human annotations as opposed to O(N^2) pairwise judgments. This has important implications both for ML practitioners interested in enriching their datasets with similarity judgments (e.g., for contrastive training) as well as for cognitive scientists interested in studying naturalistic representations.
> 2. Developing STEP-tag, an adaptive paradigm for tag mining which can be used to mine high quality text descriptors with potential applications for low-resource languages.
> 3. Evaluating and comparing our methods against 600+ publicly available pre-trained domain-specific models which cover state-of-the-art models from the literature over the last few years and serve as a baseline against the novel language based techniques we proposed.
> 4. Publicly releasing all collected datasets which comprise over 200,000 human judgments across three different modalities.
> 5. Providing a comprehensive guide for ML practitioners and scientists interested in collecting similarity judgments at scale (Figure 5).
> We now reiterate these contributions in the Discussion and Conclusion Section for clarity.
>
> ### Reviewer hUBS - Clarifying baseline evaluations
> *“No baselines are compared to the proposed method in the experiment section.”*
>
> **Response**: We strongly disagree. We have compared our text-based methods against more than 600 state-of-the-art publicly available DNN model embeddings across three modalities. As pointed out by other reviewers, this is one of the strengths of our work. While in the initial submission we did not explicitly call these models “baselines”, they definitely serve as such. Note also that this has been established as the alternative method advocated in the literature for predicting human similarity judgments (Peterson et al., 2018), making it appropriate as a baseline. To improve clarity, we now denote those models explicitly as ‘baselines’ in the figures (for example, Figure 3 DNN models that are depicted in blue are now labeled "DNN (baseline)").

---

> > ### Author Response · Authors · 2022-12-09
> > **Response to Reviewer hUBS - follow up**
> >
> > We just want to make sure the reviewer read our response. Specifically, regarding the fact that our paper actually includes substantial baseline comparisons.  We compared our methods against over 600 baseline models (which the other reviewers mentioned as one of the major strengths of our paper). To clarify this issue we updated the manuscript to further emphasize the baseline models and noted this in our reply (for example updated Figure 3 light blue models).

---

### Official Review · Reviewer_ZARH · 2022-10-26

**Confidence:** 3
**Correctness:** 4
**Technical Novelty And Significance:** 3
**Empirical Novelty And Significance:** 4
**Recommendation:** 10

**Clarity, Quality, Novelty And Reproducibility:**

-	Quality
Good. Issues for human judgment cost have been solved with the proposed idea.


-	Clarity
Good. In the experiment, model they used is sufficient.


-	Originality
Good. The idea of methods themselves are not very novel. However, I believe the exhaustive amount of examination is valuable, which increase the originality, thus it should be highly evaluated.


**Strength And Weaknesses:**

[Strength]
The concept of the paper is nicely presented for the next AI abundant society where demand for human judgments is more increased. They proposed the novel similarity approximation method with a smaller number of judgements.
They provide the evaluation data in the supplemental material, which is beneficial.
Overall, I believe what the authors done is quite systematic and very beneficial for the future refinement of the DNN model.

[Weakness]

-	Representational similarity
For beginners, the word of representation similarity is not easy to follow. At least, authors should define the terminology in the introduction.


-	The title
In particular, the linkage between title and abstract seems not clear at the first glance.
Please consider to update the title and abstract so that readers can easily understand the topic theme.


-	Is English best?
There are many languages. In my opinion, I view that authors selected English language to represent the basis of all things in nature. Why English(e.g., language system such as polysemous, grammar, dataset availability…)? Justification of English employment would make the paper better.

Related to above topic, the paper report “N = 1,492 US participants for the new behavioral experiments reported in this paper.” Are they all Americans and share the same contexts? (e.g., cultures…)

-	Crowd scouring
How long does it take to STEP-Tag? It is good that measure actual time that participants consumed. Comparing with that with the time used for writing captions may strengthen your claim.

-	Discussion
I feel strange because the paper end with “Discussion”. Please consider adding conclusion section or rename the last section as “Discussion and Conclusion”.


**Summary Of The Paper:**

To approximate human judgements without increasing cost and losing reliability, the authors proposed propose a new class of similarity approximation method based on language, namely captions and stacked words. Their method achieved good approximation with only O(N) computation cost. They also developed the novel overall flow to compute representational similarity. They also claim they are going to make the human judgement dataset publicly available.

**Summary Of The Review:**

I think this paper should be accepted for the “Human like” evaluation point of view as DNN becomes more sophisticated. I believe their finding is also beneficial for the community.

---

> ### Author Response · Authors · 2022-11-18
> **# Response to Reviewer ZARH**
>
> We are truly grateful for the reviewer’s favorable evaluation of our work and for appreciating its relevance to the community! In what follows we follow up on the comments raised in the review.
>
> ### Reviewer ZARH - Data availability
> *“They also claim they are going to make the human judgement dataset publicly available.”*
>
> **Response**: we indeed have released all data and code in the following OSF repository: https://osf.io/kzbr5/?view_only=3dea58e008ce41c290ef0f374bdbf444. In addition, to enhance transparency and accessibility we implemented an interactive guide for exploring our data which can be found under this link: https://words-are-all-you-need.s3.amazonaws.com/index.html.
>
> ### Reviewer ZARH - Abstract and title
> *“For beginners, the word of representation similarity is not easy to follow. At least, authors should define the terminology in the introduction.”*
>
> **Response**: we have updated the introduction to clarify the terminology:
> “as exemplified by the rich literature on the representational similarity between humans and machines [...] whereby similarity patterns of brain activity are compared to those arising from a model of interest.”
>
> *“In particular, the linkage between title and abstract seems not clear at the first glance. Please consider to update the title and abstract so that readers can easily understand the topic theme.”*
>
> **Response**: OpenReview does not allow us to update the title on this page at this stage. However, we have updated the title in the manuscript to “Words are all you need? Language as an approximation for human similarity judgments”.
>
> ### Reviewer ZARH - Use of English for Prediction
> *“There are many languages. In my opinion, I view that authors selected English language to represent the basis of all things in nature. Why English(e.g., language system such as polysemous, grammar, dataset availability…)? Justification of English employment would make the paper better.”*
>
> **Response**: We completely agree with the reviewer that this is an important point to discuss. In fact, we are now working on a follow-up paper expanding this work to other languages and cultures, and both STEP-tag and the embedding-free word frequency methods discussed in the paper were introduced with the goal in mind of applying our approach to low-resource languages. We have updated the discussion section of the paper to highlight this limitation of the current work and our ongoing work to address it:
>
> >“Another limitation of our work is the fact that we were restricted to English text data and US participants. On the positive side, however, we believe that our approach and proposed methods (especially STEP-tag and the word-frequency methods) pave the way for the study of cross-cultural variation of human semantic representations by providing efficient tools for crowdsourcing high-quality semantic descriptors across languages. This is particularly relevant for low-resource languages, where our tag-mining techniques can work even with the absence of pre-trained ML models (Thompson et al., 2020; Barret, 2020). We are currently expanding our work to include more languages and cultures.”
>
> *“Related to above topic, the paper report “N = 1,492 US participants for the new behavioral experiments reported in this paper.” Are they all Americans and share the same contexts? (e.g., cultures…)”*
>
> **Response**: As noted in the paper (Section 2.2.1), we required that participants on MTurk reside in the United States and that they successfully pass an English proficiency test (LexTale; Lemhofer & Broersma, 2012). We did not collect information about their cultural background. We are currently expanding our research to include cross-cultural variations.
>
> *“How long does it take to STEP-Tag? It is good that measure actual time that participants consumed. Comparing with that with the time used for writing captions may strengthen your claim.”*
>
> **Response**: For STEP-tag the median participant time spent per stimulus (i.e., running at least 10 iterations of STEP-tag involving >= 10 participants) was 230 seconds for audio and 264 seconds for videos. For captions (i.e., collecting ~10 free-text captions per stimulus), it was 187 seconds for audio and 291 seconds for video (we did not collect the timings for the image modality). We see that both methods consume roughly similar amounts of time, which is desirable as our analysis suggests that in some domains (e.g., video) tags yield the best results whereas in others (e.g. audio) captions do. We now include these time estimates in the paper (see Supplementary Section B.7 and Table 3).
>
> *“I feel strange because the paper end with “Discussion”. Please consider adding conclusion section or rename the last section as “Discussion and Conclusion”.”*
>
> **Response**: We have renamed the last section to “Discussion and Conclusion”.

---

### Author Response · Authors · 2022-11-18
**Summary of Responses**

We would like to thank the reviewers for their detailed comments and for appreciating the relevance of our work and the scale of our benchmarking experiments. We have addressed all the issues raised by the reviewers, answered all comments in OpenReview, and revised the manuscript accordingly. We are especially grateful for the suggestions the reviewers made for additional follow-up experiments and analysis, which we have fully implemented and added to the text. We summarize these as follows:
1. We now include an analysis of supervised methods whereby embeddings are fine-tuned based on a small number of similarity judgments as proposed by previous literature (Peterson et al., 2018).
2. We have added a new O(1) method which uses image-captioning models to caption images. Similarity can then be predicted by an LLM without needing to have humans caption the images. Unfortunately, this method did not perform well.
3. We converted tags into captions using a simple linearization scheme (i.e. “This is an image of tag1, tag2, …”), and tested these captions with our best-performing LLM, to compare relative contributions of type of text used and choice of model.
4. We used GPT3 in a text-completion setup to directly predict similarity without needing to extract embeddings. To do this, we created prompts from pairs of tag data, and then asked GPT-3 to rate their similarity. Even without sophisticated prompt engineering, this new O(N) method yields excellent results.
5. We added an analysis of the time taken for collecting data, and found that captions and tags take comparable amounts of time to collect, which is desirable as our analysis suggests that in some domains (e.g., video) tags yield the best results whereas in others (e.g. audio) captions do.

We believe that these suggestions and the subsequent revisions have substantially improved our work.

---

### Decision · Program_Chairs · 2023-01-20

**Decision:**

Accept: poster

**Justification For Why Not Higher Score:**

The paper extensively evaluates embeddings of pre-trained deep neural networks in approximating human similarity judgement of image, audio, and video pairs. It empirically shows that embeddings extracted by the language models on tags of samples better approximate human similarity judgements, which can be further improved by stacking embeddings of language models and pre-trained deep neural networks. As such, it can be a good poster paper.

**Justification For Why Not Lower Score:**

The paper received good scores from all reviewers, and the authors did a good job in addressing the reviewers' comments.

**Metareview: Summary, Strengths And Weaknesses:**

The paper extensively evaluates embeddings of pre-trained deep neural networks in approximating human similarity judgement of image, audio, and video pairs. It empirically shows that embeddings extracted by the language models on tags of samples better approximate human similarity judgements, which can be further improved by stacking embeddings of language models and pre-trained deep neural networks. This is an interesting finding. The paper also introduce new tagged datasets of image, audio, and video pairs. The manuscript is well-written, and the authors did a good job addressing the reviewers' comments in the rebuttal.

**Note From Pc:**

if the above contains the word "oral" or "spotlight" please see: "oral" presentation means -> notable-top-5% and "spotlight" means -> notable-top-25%. As stated in our emails, we are disassociating presentation type from AC recommendations